# Triple oxygen isotope systematics of evaporation and mixing processes in a dynamic desert lake system

Claudia Voigt[1,2], Daniel Herwartz[1], Cristina Dorador[3], Michael Staubwasser[1]

[1]Institute of Geology and Mineralogy, University of Cologne, Zülpicher Str. 49b, 50674 Cologne, Germany
[2]Aix Marseille University, Centre National de la Recherche Scientifique (CNRS), CEREGE, Aix-en-Provence, France
[3]Centro de Biotecnología, Universidad de Antofagasta, Angamos 601, 1270300 Antofagasta, Chile

*Correspondence to:* Claudia Voigt (voigt@cerege.fr)

**Abstract.**

This study investigates the combined hydrogen-deuterium and triple oxygen isotope hydrology of the Salar del Huasco, an
endorheic salt flat with shallow lakes at its centre that is located on the Altiplano Plateau, N-Chile. This lacustrine system is hydrologically dynamic and complex because it receives inflow from multiple surface and groundwater sources. It undergoes seasonal flooding, followed by rapid shrinking of the water body at the prevailing arid climate with very high evaporation rates. At any given point in time, ponds, lakes, and recharge sources capture a large range of evaporation degrees. Samples taken between 2017 and 2019 show a range of $\delta^{18}O$ between -13.3 and 14.5 ‰, d-excess between 7 and -100 ‰, and $^{17}O$-excess
between 19 and -108 per meg. A pan evaporation experiment conducted on-site was used to derive the turbulence coefficient of the Craig-Gordon isotope evaporation model for the local wind regime. This, along with sampling of atmospheric vapour at the salar (-21.0 ± 3.3 ‰ for $\delta^{18}O$, 34 ± 6 ‰ for d-excess and 23 ± 9 per meg for $^{17}O$-excess) enabled the accurate reproduction of measured ponds and lake isotope data by the Craig-Gordon model. In contrast to classic $\delta^2H$-$\delta^{18}O$ studies, the $^{17}O$-excess data not only allow to distinguish two different types of evaporation – evaporation with and without recharge – but
also to identify mixing processes between evaporated lake water and fresh flood water. Multiple generations of infiltration events can also be inferred from the triple oxygen isotope composition of inflow water, indicating mixing of sources with different evaporation histories. These processes cannot be resolved using classic $\delta^2H$-$\delta^{18}O$ data alone. Adding triple oxygen isotope measurements to isotope hydrology studies may therefore significantly improve the accuracy of a lake's hydrological balance – i.e. the evaporation-to-inflow ratio (E/I) – estimated by water isotope data and application of the Craig-Gordon
isotope evaporation model.

## 1 Introduction

The majority of water in the hydrologic cycle on Earth represents a dynamic isotopic equilibrium between evaporation from the ocean, precipitation, and continental runoff, defining a linear relationship between $\delta^{17}O$ and $\delta^{18}O$ or $\delta^2H$ and $\delta^{18}O$ – i.e. the Global Meteoric Water Line (GMWL; Clark and Fritz, 1997; Luz and Barkan, 2010). However, a number of continental water
reservoirs, e.g. lakes in semi-arid or arid environments, may deviate from that state as a result of evaporation – which imparts

an enhanced kinetic isotope effect on these waters – or because of mixing with pre-evaporated water. Such deviations are described by the secondary parameters $^{17}$O-excess [$= \delta'^{17}O-0.528\cdot\delta'^{18}O$ with $\delta'=1000\cdot(\delta/1000+1)$] and d-excess [$= \delta^2H-8\cdot\delta^{18}O$], which assume values of 33 per meg and 10 ‰, respectively, for waters plotting on the GMWL. The progress of evaporation causes a systematic decrease of $^{17}$O-excess and d-excess that is principally predictable by the Craig-Gordon

(C-G) isotope evaporation model. The C-G model is the foundation of assessing the hydrological balance, i.e. the evaporation-to-inflow ratio (E/I), of lakes using water isotope measurements. The model variables of the C-G equation – relative humidity, temperature, the isotopic composition of atmospheric vapour, continuous groundwater recharge, wind turbulence above the water surface, and salinity – depend on local climatic conditions and require quantification in order to apply the model (Gonfiantini, 1986; Horita et al., 2008; Gonfiantini et al., 2018). Of these variables, wind turbulence remains beyond

determination by direct measurement in the field. Importantly, the mixing of different lake water sources, particularly if these represent precipitation from different seasons or different evaporation histories, presents a significant challenge in deriving the hydrologic balance of lakes from water isotopes in the C-G model that cannot be resolved using classic $^2H/^1H$ and $^{18}O/^{16}O$ data (Gonfiantini, 1986). Without the means to address wind turbulence and mixing of sources directly, isotope hydrology cannot provide an unambiguous estimate of a lake's hydrological balance.

The analysis of the $^{17}O/^{16}O$ ratio in H$_2$O in addition to the commonly investigated $^2H/^1H$ and $^{18}O/^{16}O$ ratios has become an increasingly applied tool in hydrogeological research, because of some favourable properties of the triple oxygen isotope system. For example, while the secondarily derived d-excess parameter is largely dependent on relative humidity and temperature, the $^{17}$O-excess parameter has shown to be temperature insensitive in the natural range between 11-41°C (Barkan and Luz, 2005; Cao and Liu, 2011). Also, the effect of high salinity on the evaporation trajectory of the C-G model is less

significant (Surma et al., 2018). Hence, the combined analysis of $\delta^{17}O$, $\delta^{18}O$ and $\delta^2H$ provides information on fundamental processes of the hydrological cycle such as humidity, moisture sources, evaporation conditions and mixing, which would be masked by temperature variability if $\delta^{18}O$ and $\delta^2H$ were analysed alone (Angert et al., 2004; Barkan and Luz, 2005, 2007; Landais et al., 2006, 2008, 2010; Surma et al., 2015, 2018; Herwartz et al., 2017). As such, the general potential of the $^{17}$O-excess parameter as a tool to quantitatively reconstruct paleo-humidity from plant silica particles (Alexandre et al., 2018,

2019) and lake sediments (Evans et al., 2018; Gázquez et al., 2018) could be demonstrated.

The seasonal dynamics in climate variables, flooding, and their respective effect of mixing on the triple oxygen isotope composition of lakes have not yet been investigated. The present study's objectives are 1) test the potential of triple oxygen isotope analyses to resolve the fundamental hydrologic processes of evaporation, recharge and mixing of sources that cannot be resolved by the classical $\delta^2H$-$\delta^{18}O$ analyses; 2) test the robustness of the C-G model as a reliable monitoring tool for the

lake hydrologic balance in a hydrologically complex and seasonally dynamic desert lake system; and 3) demonstrate the potential of triple oxygen isotope analyses to derive the hydrological balance of lakes from water isotope and climate monitoring. For this purpose, we measured the isotopic data from springs, ponds and lake water samples from the Salar del Huasco, N-Chile, covering a 300 ‰ salinity range, and compared them with evaporation trajectories modelled for local seasonal conditions. All variables of the C-G model were constrained by meteorological data from an on-site weather station,

a pan evaporation experiment, isotopic measurements of atmospheric water vapour samples and inflowing water sources. Due to the extreme dynamics in climatic and hydrological parameters, the Salar del Huasco provides an ideal environment to test the potential of triple oxygen isotope analyses in resolving different hydrological processes like permanent recharge and highly episodic mixing with flood water and to identify changes in the hydrological balance of lakes.

## 2 Study Area

The Salar del Huasco is an endorheic salt flat located in the south of a longitudinal volcano-tectonic depression at the western margin of the Chilean Altiplano at about 3770 m (Fig. 1). It covers an area of about 50 km² but only 5 % of the surface is permanently covered by water (Risacher et al., 2003). The hydrological balance of the salar is controlled by a shallow groundwater table, perennial streams, rare and highly seasonal precipitation, and episodic injection of runoff water after heavy rainfalls or snowmelt. The environment is characterized by exceptionally high evaporation rates and high variability in relative

humidity and temperature throughout the year with considerable diurnal amplitude.

An on-site weather station (20°15.42'S 68°52.38'W) is operated by the Centro de Estudios Avanzados en Zonas Áridas (CEAZA) at the north-western margin of the salar since September 2015 (CEAZA, 2020). Over this period, mean annual relative humidity and temperature are 41 % and 5°C, respectively. Both relative humidity and temperature are highly variable throughout the year with mean seasonal values of 54 % and 8°C in austral summer (Dec–Mar) and 33 % and 2°C in austral

winter (Jun–Sep). Day-night fluctuations range between 0 and 99 %, and from -19°C to 23°C. Mean annual wind speed is approximately 2.5 m·s$^{-1}$, most times coming from S to SW direction. Typically, it is calm in the morning (< 1 m·s$^{-1}$), but very windy in the afternoon with 5 m·s$^{-1}$ average wind speed and gusts up to 24 m·s$^{-1}$.

Long-term mean annual precipitation is 150 mm·a$^{-1}$, but interannual variability is high due to their dependence on wind patterns (Garreaud and Aceituno, 2001; Risacher et al., 2003). Precipitation occurs mainly in austral summer (Dec–Mar) and is related

to easterly airflow from the Atlantic Ocean and the Amazon Basin (e.g. Aravena et al., 1999; Garreaud et al., 2003; Houston, 2006). In contrast, less frequent winter rains and snow are associated with the interaction of cold air masses from the Pacific with tropical air masses from the Amazon Basin (Aravena et al., 1999). The intensity of convective storms and variability of the continental effect in the Amazon Basin lead to significant isotopic variability in summer rains (Aravena et al., 1999). In contrast, winter rains comprise generally higher $\delta^{18}$O values and are isotopically less variable (Aravena et al., 1999). This may

be attributed to lower variability in the contribution of pre-evaporated water from the Amazon Basin in the dry season and the non-convective nature of winter storms.

The hydrogeological system of the Salar del Huasco consists of three aquifers, where the upper and the intermediate aquifer are located in the basin's sediment infill, separated by a thin aquitard, and the lower aquifer is formed in volcanic bedrock (Acosta and Custodio, 2008). The upper aquifer is recharged by the Collacagua river that drains the northern part of the basin

and completely infiltrates 10 km before reaching the salar (Acosta and Custodio, 2008). In periods of heavy rainfall, the river directly flows into the salar leading to widespread flooding of the northern area (Fig. 2e). Several springs and creeks around

the salar and the shallow groundwater table contribute to the formation of perennial lakes. Furthermore, ephemeral ponds emerge episodically due to precipitation, surface and subsurface runoff and a rising groundwater table in the rainy season (austral summer). Episodic precipitation, surface runoff, ephemeral flooding and variable contributions from different groundwater and stream sources result in a highly dynamic system with strongly fluctuating lake level.

## 3 Sampling

Natural water samples from the Salar del Huasco were taken during field campaigns in September 2017, September 2018, and March 2019 (Fig. 2 and Table S2). The sample set includes springs, perennial lakes (*Laguna Grande* in the W, and *Laguna Jasure* in the SE) and ephemeral ponds of which several were apparently recharged and some apparently stagnant. Lakes and ponds are all very shallow (5–30 cm). Samples were taken from the water surface, and temperature, pH and conductivity were measured on-site using a digital precision meter *Multi 3620 IDS*.

Based on satellite images and field observations, five major hydrological subsystems could be identified (Fig. 2). The *Laguna Grande* in the western part of the salt flat is intermediate saline with values varying seasonally between approximately 20 and 35 $g·l^{-1}$.

A channel originating from the southern end of this lagoon extends along the southern margin of the salt flat. This channel contributes to flooding of the southern area during the rainy season, while lowering of the water table over the course of the year leads to isolation of ponds (Fig. 2e and g). The large range in salinity ($1 – >100$ $g·l^{-1}$) observed in shallow lakes and ponds at the south to south-western margin of the salar may indicate that their hydrological balance alternates between recharge and isolation.

In the south-eastern area, a perennial spring fed a chain of lakes with generally low salinity of $< 2$ $g·l^{-1}$. Some ponds were visibly connected by streams and creeks. Others located closer to the eastern margin of the salt flat, were topographically elevated and therefore isolated from the main south-eastern inflow. Satellite images indicate that ponds closer to the salar's centre may represent a mixture of source water from the south-eastern inflow and a further subsystem in the north (Fig. 2e).

The northern subsystem is probably fed by subsurface inflow of the Collacagua river. Its area may become widely flooded in austral summer (Fig. 2e) but dries up rapidly after the rainy season (Fig. 2g). Ponds from the northern area sampled in 09/17 were characterized by an extreme salinity gradient from 2 to 343 $g·l^{-1}$, broadly decreasing from east to west (Table S2). These ponds were often salt-encrusted and showed black, sulphur-reducing microbial mats at the bottom. In 03/19, this entire area was covered by a large low-salinity lake.

In the north-western area, a spring originating from a small, vegetated hill fed a series of ponds with salinities $< 1$ $g·l^{-1}$. An adjacent chain of ponds sampled in 09/17 about 750 m to the south-east with salinities between 1 and 6 $g·l^{-1}$ was not visibly connected to this spring and must have been sustained by the shallow groundwater table. Three topographically elevated ponds close to the margin had higher salinity of 8 to 40 $g·l^{-1}$. The area around these ponds was more vegetated, indicating that they

were older, i.e. represented an earlier flooding but became isolated from recharge. Despite extensive rainfall-induced-flooding in February 2019, the north-western part of the salt flat was almost dry in March 2019.

Besides water samples from surface waters, we sampled atmospheric vapour using a Stirling cooler cycle system (*Le-Tehnika*, Kranj, Slovenia) built after Peters and Yakir (2010). Sampling was carried out on three days during field campaigns in 09/17 and 03/19 (Table 1). Atmospheric vapour was sampled at ground level. Water vapour was captured by streaming air with a flow rate of 600-800 ml·min$^{-1}$ through a 4 ml vial, which was attached to the cold finger of the Stirling cooler and insulated. Extremely low absolute humidity values at the Salar del Huasco required sampling over several hours to yield 0.1-0.2 µl of

atmospheric vapour.

Additionally, pan evaporation experiments with 600 ml, 800 ml, and 1000 ml of fresh water (0.805 mS·cm$^{-1}$) filled in stainless steel evaporation pans (Ø 20 cm) were carried out on-site over a period of three days (20°15.4'S 68°52.4'E). The rationale behind this experimental design was to account for the extreme diurnal and significant average day-to-day variability of all variables in the C-G model by letting evaporation progress over several days. By varying the initial volume, we were able to

achieve a wide spread of fractional evaporative water loss. Samples were taken every day around 18:00 and, additionally after 13:00 on the third day. The 600 ml pan dried up before the end of the experiment so that no sample could be taken in the evening of the third day. Air temperature, relative humidity, and wind speed were monitored locally at the experiment at about 1.5 m above ground using a *Kestrel 5500 weather meter* (Fig. S2). Mean temperature and relative humidity were 4°C and 35 % over the whole period of the experiment. As temperatures dropped below 0°C in the night, a considerable fraction, if not all,

of the water in the pans froze during the night. This led to the formation of either a solid ice block or a thick ice layer above the remaining liquid. Consequently, evaporation from pans was mostly restricted to daytime when samples were thawed. The effective evaporation time interval was assumed to correspond to the period when T > 0°C, resulting in average air temperature and relative humidity values of 10°C and 22 % (Fig. S2). Water temperatures measured during sampling were found to be up to 5°C lower than ambient air temperatures. However, during midday, temperatures of water may exceed air temperatures by

several degrees due to solar heating of the pans. Winds were very strong between 12:00 and 19:00 with an average wind speed of 5 m·s$^{-1}$ and gusts up to 14 m·s$^{-1}$ coming from S to W direction.

## 4 Methods

### 4.1 Isotope and chemical analyses

The hydrogen and triple oxygen isotope composition of water samples were analysed by isotope ratio mass spectrometry

(IRMS). Complementary concentration data of $Na^+$, $K^+$, $Ca^{2+}$, $Mg^{2+}$, $Cl^-$, and $SO_4^{2-}$ in natural samples were determined by ICP-OES (Table S2). A short discussion of the chemical data is provided in the supplement (T2, Fig S7).

Hydrogen isotope ratios are measured by continuous-flow IRMS of $H_2$. Water samples are injected in a silicon carbide reactor (*Heka-Tech*, Wegberg, Germany) that is filled with glassy carbon and heated to 1550°C, where they are reduced to $H_2$ and CO. The produced gases are carried in a helium gas stream (100-130 ml·min$^{-1}$), separated by gas chromatography (GC) and finally

introduced in a *Thermo Scientific MAT 253* mass spectrometer for hydrogen isotope analysis. The long-term external reproducibility (SD) is 0.9 ‰ and 1 ‰ for $\delta^2H$ and d-excess, respectively.

For triple oxygen isotope analysis, water samples are fluorinated, followed by dual-inlet IRMS of $O_2$. The method is described in detail in Surma et al. (2015) and Herwartz et al. (2017). In brief, 2.8 µl of water are injected in a heated $CoF_3$ reactor (370°C) that is continuously flushed with helium (30 ml·min⁻¹). The produced oxygen gas is cryogenically purified and trapped in one of twelve sample tubes of a manifold. The manifold is connected to a *Thermo Scientific MAT 253* for dual-inlet IRMS analysis. The long-term external reproducibility (SD) is 0.12 ‰, 0.25 ‰ and 8 per meg for $\delta^{17}O$, $\delta^{18}O$ and $^{17}O$-excess, respectively. All isotope data herein are reported on SMOW–SLAP scale (Schoenemann et al., 2013). The scale is usually contracted using the setup described herein. This can partly be attributed to blank contribution (Herwartz et al., 2017). We observed an increase in scale contraction over the usage period of a $CoF_3$ reactor filling and a reduction of precision and accuracy of isotopic data, indicating that the blank contribution increases with time. To account for this effect, SMOW–SLAP scaling was performed daily using internal standards. Isotope measurements with anomalous high scaling factors or standard deviations were discarded.

**4.2 The Craig-Gordon isotope evaporation model at the Salar del Huasco**

The Craig-Gordon (C-G) isotope evaporation model forms the basis for numerous models describing the isotopic composition of natural lakes (e.g. Gonfiantini, 1986; Gat and Bowser, 1991; Horita et al., 2008). There are two principal evaporation scenarios – one without recharge (hereinafter termed 'simple evaporation') and one with continuous recharge (hereinafter termed 'recharge evaporation') (Craig and Gordon, 1965; Criss, 1999; Horita et al., 2008). In the case of simple evaporation, the isotopic composition of the lake is only controlled by the degree of evaporation (Fig. 3; Gonfiantini et al., 2018):

$$R_W = f^B \cdot \left( R_{WI} - \frac{A}{B} \cdot R_V \right) + \frac{A}{B} \cdot R_V$$

where $R_{WI}$ denotes the initial isotopic ratio ($^2H/^1H$, $^{17}O/^{16}O$, $^{18}O/^{16}O$) in the water body, $R_V$ is the isotopic ratio in atmospheric vapour, and f is the fraction of residual water. Note that R can be inferred from the δ-notation by R = δ/1000+1. The parameters A and B describe the isotopic fractionation associated with evaporation in dependence on the relative humidity, h, normalized to the water surface temperature:

$$A = -\frac{h}{\alpha_{\text{diff, l}-v} \cdot (1-h)}$$

$$B = \frac{1}{\alpha_{\text{eq, l}-v} \cdot \alpha_{\text{diff, l}-v} \cdot (1-h)} - 1$$

Diffusive ($\alpha_{\text{diff, l}-v}$) and equilibrium ($\alpha_{\text{eq, l}-v}$) isotope fractionation cause a systematic increase in $\delta^{18}O$ and decrease in d-excess and $^{17}O$-excess with progressive evaporation. A detailed description of all variables used herein is given in the supplement (T1, Table S1).

In the case of recharge evaporation, the isotopic effect of continuous inflow is accounted for by the evaporation-to-inflow-ratio (E/I) (Fig. 3; Criss, 1999):

$$R_{WS} = \frac{\alpha_{eq,\,l-v} \cdot \alpha_{diff\,\,l-v} \cdot (1-h) \cdot R_{WI} + \alpha_{eq,\,l-v} \cdot h \cdot E/I \cdot R_V}{E/I + \alpha_{eq,\,l-v} \cdot \alpha_{diff,\,l-v} \cdot (1-h) \cdot (1-E/I)}$$

Here, $R_{WI}$ refers to the isotopic ratio in the inflowing water. Under steady-state conditions $E/I \leq 1$, whereas the lake begins to desiccate when evaporation exceeds the inflow ($E/I > 1$).

The effect of wind turbulence – which leads to variable proportions of diffusive and turbulent isotope fractionation in dependence on wind speed – is accounted for by inserting a correcting exponent to the diffusive fractionation factor, $\alpha_{diff,\,l-v}^{n}$ (Dongmann et al., 1974). The turbulence coefficient n can vary between 0 (fully turbulent atmosphere) and 1 (calm atmosphere), but typically assumes values of $n \geq 0.5$ under natural conditions (Gonfiantini, 1986; Mathieu and Bariac, 1996; Surma et al., 2018). Recent laboratory experiments suggest that part of the evaporating water is removed by spray and microdroplet vaporization without any isotope fractionation when wind speed exceeds $\sim 0.5$ m·s$^{-1}$, which may require a modification of the above described C-G approach (Gonfiantini et al., 2020). We evaluated the effect of a possible partial evaporative water loss without fractionation on the simple evaporation trajectory by introducing a 'virtual outlet' $f_{out}$ in the isotopic mass balance equation, which finally results in a modification of the parameter B:

$$B_{out} = \frac{1 - f_{out}}{\alpha_{eq,\,l-v} \cdot \alpha_{diff,\,l-v}^{n} \cdot (1-h)} + \frac{f_{out}}{(1-h)} - 1$$

Salinity affects isotope activities and increases fluid viscosity thereby decreasing the vapour pressure above the water body. In the C-G model, this may be accounted for by correcting equilibrium fractionation factors for the classic salt effect (Horita, 1989, 2005; Horita et al., 1993) and using effective rather than actual relative humidity. The salinity effect requires consideration within the $\delta^2$H-$\delta^{18}$O system at salinities > 100 g·l$^{-1}$ and is much less significant for the $\delta^{17}$O-$\delta^{18}$O system (Sofer and Gat, 1972, 1975; Horita, 1989, 2005; Surma et al., 2018). To avoid unnecessary complication of the more principal approach of this study, salinity effects were neglected in our model calculations.

The C-G model does not account for mixing processes, e.g. as a result of flooding or snowmelt, but can be used to calculate such effect by applying mass balance: $\delta X_{mix} = f \cdot \delta X_1 + (1-f) \cdot \delta X_2$, where $\delta X_1$ and $\delta X_2$ represent the isotopic composition of two different water sources and $\delta X_{mix}$ is the isotopic composition of the resulting mixed water body with X denoting $^{18}$O, $^{17}$O, or $^2$H, respectively. Mixing curves are inversely shaped to the evaporation trajectories in triple oxygen isotope space (Fig. 3). At the Salar del Huasco mixing may occur episodically due to flooding and fluctuations in the groundwater level as detectable on satellite images (Fig. 2). Such mixing processes are likely transient due to the rarity of flooding events. In our model approach, we assume that the isotopic composition of the admixed water is similar to that of spring water, which approximates the isotopic composition of groundwater reasonably well (Uribe et al., 2015).

Simple evaporation, recharge evaporation and mixing are principally well resolvable in triple oxygen isotope space, whereas in $\delta^2$H-$\delta^{18}$O space, they tend to merge within data uncertainty (Fig. 3). All three trajectories can be expected in a dynamic arid hydrological setting such as the Salar del Huasco. Water affected exclusively by evaporation must progress along either of the

two principal evaporation trajectories defined by the C-G model, depending on the recharge conditions. On the other hand, episodic flooding events initiate mixing leading to transient deviations from these general evaporation trends. The relative isotopic difference between inflowing water ($\delta_{WI}$) and atmospheric vapour ($\delta_V$) mainly determines the resolution of evaporation trajectories for different relative humidity in the triple oxygen isotope plot (Surma et al., 2018). For the given boundary conditions at the Salar del Huasco, evaporation trajectories show low sensitivity to changes in relative humidity, temperature, and the turbulence coefficient in the diagram of $^{17}$O-excess over $\delta^{18}$O (Fig. S3). Thus, seasonal and diurnal variability in these climate variables should have a low impact on the triple oxygen isotope composition of lakes and ponds at the Salar del Huasco, which is favourable for the purpose of this study to resolve different hydrological processes of evaporation, recharge, and mixing.

### 4.3 The isotopic composition of atmospheric vapour

A few atmospheric vapour samples were extracted on-site using a battery-powered Stirling cooler (Peters and Yakir, 2010). Due to the low relative humidity at the study site, extractions lasted 2-3 hours and only three samples could be taken.

In the absence of sufficiently abundant direct measurements, the isotopic composition of atmospheric vapour may be inferred from precipitation data, e.g. derived from the Online Isotopes in Precipitation Calculator (OIPC) (Bowen, 2020), assuming isotopic equilibrium [$\delta X_V = (\delta X_P + 1000)/\alpha_{eq,\,l-v} - 1000$, where $\delta X_P$ refers to the isotopic composition of precipitation and X denotes $^{18}$O, $^{17}$O, or $^2$H, respectively]. However, the applicability of the equilibrium assumption is questionable for the Salar del Huasco, where precipitation mostly occurs in rare thunderstorm events originating from easterly sources. More importantly, rain samples cannot be representative for the local vapour because throughout the majority of the year air masses originate from westerly Pacific sources, which generally do not lead to precipitation (Garreaud et al. 2003). To confirm the predominant western origin of vapour for the time of this study, the HYSPLIT Lagrangian model (Stein et al., 2015) was used to calculate seven-day air mass back-trajectories in daily resolution (12:00) for the month prior to each of the sampling campaigns for ground level (3800 m above sea level (asl)) and 1500 m above ground (5300 m asl) (c.f. Aravena et al., 1999). Additionally, air mass back-trajectories were modelled in hourly resolution for the time of vapour sampling to confirm the Pacific origin of sampled vapour. Because direct vapour extraction in the arid environment requires several hours of attendance, we were only able to extract a limited number of point-in-time samples. To augment the scarce data base, we verified our $\delta^{18}$O$_v$ indirectly from the on-site evaporation experiments.

### 4.4 Isotope turnover time of ponds

The isotopic composition of a pond represents an integrated signal over the turnover time of the isotopes, which needs to be estimated to accurately apply the C-G model. This isotope turnover time is specific for each pond and depends on its surface-to-volume ratio, recharge, and the evaporation rate. Only very small water volumes may capture diurnal variations as shown in pan evaporation experiments (Surma et al., 2018). In contrast, deeper water bodies integrate over longer time intervals. Diurnal cycles and changing conditions over several days or weeks will be smoothed out, particularly if local conditions, e.g.

a rough wind regime, lead to a well-mixed lake. The isotope turnover time may be approximated considering the turnover rate of ponds. For a terminal lake, the turnover rate (TR) equals the vaporization rate ($\varphi_{vap}$), which is a function of the potential evaporation ($E_{pot}$) and relative humidity (h): $TR_{E/I=1} = \varphi_{vap} = E_{pot}/(1-h)$ (Gonfiantini et al., 2018). Mean annual potential evaporation at the Salar del Huasco of 2290 mm (DGA, 1987) together with the mean annual relative humidity of 40 % results in a mean turnover rate of 318 mm/month (11 mm/d). As isotopic steady-state conditions are reached asymptotically, the isotope turnover time is better approximated by the isotope 'half-turnover time': $t_{1/2} = \ln(2)/(d \cdot TR)$, with $d$ denoting the depth of the water body. Ponds in the Salar del Huasco are very shallow with depths up to 30 cm, which results in an isotope half-turnover time of < 20 days. Considering seasonal variability of the potential evaporation rates and relative humidity (150 mm/month and 33 % in austral winter, 200 mm/month and 54 % in austral summer; DGA, 1987; CEAZA, 2020), the isotope half-turnover time varies between 14 and 26 days for a 30 cm deep pond. To account for the isotope half-turnover time described above, evaporation trajectories were modelled with temperature and relative humidity values averaged over 10 or 20 days prior to sampling in March and September, respectively (Fig. 4). Diurnal variations in the evaporation rate were accounted for using daytime (6-18 h) values.

## 5 Results

### 5.1 Natural waters in the Salar del Huasco basin

Springs sampled in 09/17 comprise average values of -12.5 ± 0.6 ‰ for $\delta^{18}O$, 2 ± 3 ‰ for d-excess, and 11 ± 7 per meg for $^{17}O$-excess, which are in good agreement with published isotopic data of springs and wells in the Salar del Huasco basin (-12.56 ± 1.36 ‰ in $\delta^{18}O$ and 3.2 ± 5.7 ‰ in d-excess; data from Fritz et al., 1981; Uribe et al., 2015; Jayne et al., 2016). The springs' isotopic composition shows slight intra- and interannual variability comprising average $\delta^{18}O$, d-excess and $^{17}O$-excess values of -12.3 ± 0.5 ‰, 1 ± 2 ‰ and 6 ± 7 per meg in 09/18 and -12.6 ± 0.7 ‰, 6 ± 2 ‰, and 2 ± 6 per meg in 03/19, respectively. The Collacagua river, which was sampled at about 15 km distance from the salar, revealed similar isotopic composition with $\delta^{18}O$, d-excess and $^{17}O$-excess values of -12.4 ‰, 2 ‰ and 5 per meg in 09/18 and -12.5 ‰, 6 ‰ and 5 per meg in 03/19, respectively.

The ponds in the Salar del Huasco show typical evaporation trends of increasing $\delta^{18}O$ values with decreasing d-excess and $^{17}O$-excess values ranging from -11.2 to 14.5 ‰ in $\delta^{18}O$, from -1 to -100 ‰ in d-excess, and from 19 to -108 per meg in $^{17}O$-excess (Fig. 4). Ponds from the north-western and south-eastern subsystem are generally less enriched in $\delta^{18}O$ compared to ponds from the northern subsystem. Ponds in the south-western area span a wide range in isotopic composition from -5.5 ‰ to approximately 14.5 ‰, including the highest $\delta^{18}O$ values observed in our sample set. The permanent Laguna Grande has a composition in the intermediate range of all investigated ponds showing considerable seasonal and interannual variability between -4.2 and 3.2 ‰ in $\delta^{18}O$, -2 and -42 ‰ in d-excess, and -15 and 39 per meg in $^{17}O$-excess.

**5.2 Atmospheric vapour**

Two atmospheric vapour samples taken during the field campaign in 09/17 yield in average -19.4 ± 2.3 ‰ for $\delta^{18}O_V$, 30 ± 3 ‰ for d-excess$_V$ and 18 ± 2 per meg for $^{17}O$-excess$_V$. An additional vapour sample taken in 03/19 comprise a slightly more depleted $\delta^{18}O_V$ value of -24.3 ‰, and d-excess$_V$ and $^{17}O$-excess$_V$ values of 41 ‰ and 33 per meg, respectively. The overall average of all three measurements is -21.0 ± 3.3 ‰, 34 ± 6 ‰ and 23 ± 9 per meg for $\delta^{18}O_V$, d-excess$_V$ and $^{17}O$-excess$_V$, respectively.

Seven-day air mass back-trajectories calculated using the HYSPLIT model suggest that air masses during vapour sampling were mainly derived from Pacific sources (Fig. 5). A predominantly western origin of air masses for the month prior to sampling is confirmed by 7-day air mass back-trajectories (Fig. 5). These model results support the assumption that direct vapour measurements reflect the mean annual isotopic composition of atmospheric vapour at the Salar del Huasco.

The measured mean $\delta^{18}O_V$ value of -21.0 ± 3.3 ‰ is comparable to the mean annual $\delta^{18}O_V$ value of -21.8 ‰ calculated based on monthly precipitation data derived from the OIPC database (Bowen et al., 2005). The apparent similarity between the OIPC model and our measurements may be coincident resulting from the seasonality of precipitation sources with a major contribution of the depleted Amazon moisture source and a minor contribution of the relatively enriched winter snow moisture source (cf. Aravena et al., 1999).

**5.3 Pan evaporation experiments and the determination of the local turbulence coefficient**

Water samples from the evaporation experiments comprise increasing $\delta^{18}O$ and decreasing $^{17}O$-excess and d-excess values with increasing degree of evaporation (Fig. 6). The observed trends are consistent between experiments carried out with different initial volume. The isotopic composition of the initial water was -11.0 ‰, 4 ‰ and 16 per meg, for $\delta^{18}O$, d-excess and $^{17}O$-excess, respectively. Over the three-days experimental period, the largest enrichment in $\delta^{18}O$ ($\Delta$ = 31.4 ‰) and depletion in d-excess ($\Delta$ = -120 ‰) and $^{17}O$-excess ($\Delta$ = -155 per meg) is observed for the evaporation pan with the lowest initial volume (600 ml), which reflects the maximum fractional loss by evaporation relative to the initial volume.

The pan evaporation experiment data was used to empirically determine the turbulence coefficient n. The turbulence coefficient may be derived by the best fit of the C-G evaporation trajectory through all experimental data in plots of $\delta^{18}O$ and $\delta^2H$ over the fraction of remaining water. Alternatively, the turbulence coefficient may be determined by fitting the evaporation trajectory through the experimental data in the plot of d-excess over the residual fraction. The latter method is advantageous, because here, the evaporation trajectory is predominantly controlled by the magnitude of the turbulence coefficient (Fig. 7a), and only barely sensitive to the other variables of the C-G equation (Fig. S3). Importantly, this approach is insensitive to $\delta^{18}O_V$, the variable that is least constrained in our study (Fig. 7b).

Using model input parameters summarized in Table 2, fitting the evaporation model through all experimental data in the plot of d-excess over the fraction of remaining water implies n = 0.44 ± 0.04. This value falls within error in the lowermost range of turbulence coefficients ≥ 0.5 that are typically observed under natural conditions world-wide (Merlivat and Jouzel, 1979;

Gonfiantini, 1986; Mathieu and Bariac, 1996; Surma et al., 2018; Gázquez et al., 2018) and apparently reflects excessive evaporation during midday at higher than average turbulence caused by prevailing strong winds with gusts up to 24 m·s$^{-1}$. However, fitting the evaporation trajectory independently to $\delta^{18}O$ and $\delta^2H$ data results in a discrepancy between the individually derived turbulence coefficients. The $\delta^{18}O$ and $\delta^2H$ data of our evaporation experiment may only fit the C-G model at an unrealistically low value of n = 0.34 for $\delta^{18}O$, while no fit can be achieved for any value of $0 \leq n \leq 1$ in the diagram of $\delta^2H$ vs the residual fraction (Fig. 8). Similar discrepancies were observed in laboratory experiments at wind speeds above 0.5 m·s$^{-1}$ carried out by Gonfiantini et al. (2020).

In this particular experiment, the freezing at night-time might have biased the relationship between d-excess and the fraction of remaining water. When the ice begins to melt in the morning, a considerable fraction, if not the whole resulting water film on top of the pan's frozen surface layer may evaporate in isolation from the bulk of water underneath the ice. The fraction of total water in the pan would thus be reduced without affecting the isotopic composition of the bulk ice. Sublimation of ice at night would add to this effect. A similar effect would be introduced by microdroplet vaporization without isotope fractionation during daytime at strong winds as suggested by Gonfiantini et al. (2020). Applying a 'virtual outlet' introduced by these authors – i.e. a fraction of water that is lost without fractionation – in the isotope mass balance equation yields an excellent fit in all three plots of d-excess, $\delta^{18}O$ and $\delta^2H$ vs residual fraction for n = 0.59 ± 0.06 with 20 ±4 % loss of water without fractionation (Fig. 9). The hypothesized loss of water without isotope fractionation – either during thawing, due to sublimation or microdroplet vaporization – would have a negligible effect on isotopic data in a diagram of d-excess over $\delta^{18}O$ (Fig. 6b). At given boundary conditions, the best fit for this purely isotopic correlation is obtained for a value of n = 0.55 ± 0.09, which is within error identical with the turbulence coefficient n = 0.59 ± 0.06 derived from the virtual outlet model.

The disadvantage of the latter model is the requirement of a $\delta^{18}O_V$ value, which in this case is only poorly constrained by a few measurements, which in turn translates into a higher uncertainty. For the above calculation, $\delta^{18}O_V$ = -19.4 ± 3 ‰ was used, as derived from our own vapour measurements carried out during the period of the experiment. The obtained value of n = 0.55 ± 0.09 is in good agreement with the global range of reported turbulence coefficients (Merlivat and Jouzel, 1979; Gonfiantini, 1986; Mathieu and Bariac, 1996; Surma et al., 2018), and thus indirectly supports the accuracy of the measured $\delta^{18}O_V$ value.

In a diagram of $^{17}O$-excess vs $\delta^{18}O$, the isotopic data fall below the predicted evaporation trend, regardless which model or turbulence coefficient is used (Fig. 6a). This mismatch may be caused by two effects: 1) Partial melting during the thawing of our experiment in the morning may result in uncontrollable mixing effects that are only observable in a $^{17}O$-excess over $\delta^{18}O$ diagram (Fig S5); 2) Diurnal variations of temperature and relative humidity, generate corresponding changes of the theoretical isotopic end point of evaporation leading to a diurnal evolution of the simple evaporation trajectory (Fig. S6; Surma et al., 2018). Both effects become increasingly important, as the experiment progresses to smaller residual water volumes, consistent with our data. High-resolution sampling of similar evaporation experiments may aid in resolving these two effects in the future.

In conclusion of this experiment, we suggest that the complementary analysis of all water isotopes in an evaporation experiment principally allows the determination of the turbulence coefficient. The effect of freezing and thawing on the total water loss

was not fully resolvable from our experimental data. Hence, we used the turbulence coefficient of n = 0.55 for the following modelling calculations, which was derived from the isotopic correlation between d-excess and $\delta^{18}O$, independently from the fraction of remaining water.

## 6. Discussion: Isotopic hydrology in the Salar del Huasco Basin

### 6.1 Mixing in tributaries and groundwater aquifers

The Collacagua river and its tributaries originate from springs at different altitudes in the Salar del Huasco basin (c.f. Fig. 1) and could be expected to comprise a somewhat lower $\delta^2H$ and $\delta^{18}O$ composition than evaporated salar water due to the altitude effect (Uribe et al., 2015). Samples from the Collacagua river fall below the LMWL/GMWL (Fig. 10), reflecting a significant evaporation history. Likewise, the $\delta^{18}O$ values of springs in the Salar del Huasco basin are slightly higher than local precipitation ($\delta^{18}O$ = -17 to -13 ‰; Fritz et al., 1981; Aravena et al., 1999; Uribe et al., 2015; Scheihing et al., 2018), and the d-excess as well as [17]O-excess generally fall below the LMWL/GMWL (Fig. 10). This is a common observation for groundwater in the Atacama Desert and other desert environments (Aravena, 1995; Surma et al., 2015, 2018). Based on diagrams of d-excess over $\delta^{18}O$, this offset has been attributed to evaporation of precipitation during infiltration into the soil in previous studies (Aravena, 1995; Fig. 10b). However, all spring and river samples fall below any reasonable evaporation trend in triple oxygen isotope space (Fig. 10a). As such, evaporation along the river path or within the aquifer cannot be solely responsible for the observed isotopic composition. Additional modification of river and groundwater by mixing of different sources must be invoked to explain the triple oxygen isotope composition. Mixing most likely occurs during infiltration into the soil between precipitation and older, evaporated connate water in the vadose zone. Mixing may also take place with water adsorbed onto salts, e. g. halite (NaCl), or structurally bonded water of minerals, e.g. mirabilite ($Na_2SO_4 \cdot 10H_2O$), which we observed frequently during sampling in the Salar del Huasco environment.

### 6.2 Hydrological processes in the salar

The isotopic composition of lakes and ponds from the Salar del Huasco sampled in the period from 09/17 to 03/19 principally reflects the evaporation trend predicted by the C-G model for given boundary conditions (Table 3; Fig. 4). Most of the ponds fall on the recharge evaporation trajectory indicating that evaporation occurs while recharge takes place by surface inflow from springs, streams and creeks as well as subsurface inflow by groundwater. Ponds with high $\delta^{18}O$ values sampled in the southern/south-western area of the salar in 09/17 and 09/18 fall within the envelope spanned by the trajectories for recharge evaporation and simple evaporation at the same boundary conditions (Fig. 4a, c). These ponds may have been isolated from recharge a short time before sampling due to the general lowering of the water table during the dry season and evolved from the recharge evaporation trajectory towards the simple evaporation trajectory without a general change in climatic boundary conditions.

Other seemingly isolated ponds from the overflow region connecting the *Laguna Jasure* with the northern subsystem, sampled in 03/19, clearly fall on the simple evaporation trajectory (Fig. 4e). These ponds were probably remnants of a major flooding in the previous austral summer and cut off from recharge for long enough prior to sampling to evolve all the way from the recharge evaporation trajectory or a mixing curve (see below) to the simple evaporation trajectory. In general, both evaporation trends – recharge evaporation and simple evaporation – are well resolved in the plot of $^{17}$O-excess over $\delta^{18}$O (Fig. 4a, c, e) but are indistinguishable in a diagram of d-excess over $\delta^{18}$O (Fig. 4b, d, f).

Ponds sampled in 09/17 in the northern and eastern part of the salar fall generally below the predicted recharge evaporation trajectory, but instead within the envelope for mixing defined by the inflow and terminal lake endmembers (Fig. 4a). Thus, in addition to evaporation and recharge, mixing of younger, less evaporated floodwater and older, more evaporated lake water is at least of temporary importance. At the Salar del Huasco, the seasonality of precipitation and the impact of evaporation cause fluctuations in the groundwater table and occasional flooding. An increase in the groundwater table after heavy rainfalls or snowmelt may lead to admixture of fresh water to the pre-evaporated shallow subsurface flow or pond water. The episodic nature of precipitation should result in rather transient mixing events at the Salar del Huasco. This episodic mixing model differs from the continuous mixing hypothesis proposed by Herwartz et al. (2017) at the Salar de Llamará in the Central Depression of the Atacama Desert, where continuous admixture of isotopically light groundwater to a pre-evaporated subsurface flow may lead to systematic variations in the isotopic composition of inflowing water along the flow path.

**6.3 Impact of climatic dynamics**

Considerable seasonal and diurnal variability in environmental conditions at the Salar del Huasco impose variations in climatic boundary conditions during the isotope turnover time of a few weeks to a few months. Moreover, the observed mixing of different generations of precipitation during infiltration and possibly longer residence time of water within the aquifers feeding the spring sources of the lakes and ponds in the Salar del Huasco should result in some variability of the starting point of an evaporation trajectory. Consequently, triple oxygen isotope evaporation trajectories of the C-G model are subject to shifts over time. To evaluate the impact of such variability on the C-G model's resolving power of hydrological processes and detection of mixing from model mismatch in particular, different scenarios were simulated within the range of observed source water and climate variability. The following simulations were focussed on our main field campaign in 09/17, where mixing was identified to have affected the isotopic composition of ponds.

The effect of source variability is most visible in a broader range of evaporation trajectories in the diagram of $^{17}$O-excess over $\delta^{18}$O for through-flow ponds and lakes with E/I < 0.5 (Fig. 11a). At the Salar del Huasco, source variability may account for the offset of many of the ponds, but a number of those with seemingly high E/I ratios fall well below the simulated envelope. The impact of variability in the isotopic composition of ambient vapour ($\delta^{18}$O$_V$) that could potentially affect evaporation trajectories in the C-G model, was estimated using the standard deviation (1 SD = 3.3 ‰) of the three direct measurements distributed over the time interval of this study (Fig. 11c-d). Again, many of the ponds are plotting inside the predicted evaporation envelope defined by variability of $\delta^{18}$O$_V$ except for those ponds with high E/I that again fall outside the range. To

explain these ponds with an evaporation trajectory shifted by variability in the ambient vapour composition, $\delta^{18}O_V$ values of about -30 ‰ or even lower would be necessary. This seems to be unlikely considering our measurements (-21.0 ± 3.3 ‰). Most importantly, however, the combined $^2H/^1H$, $^{17}O/^{16}O$ and $^{18}O/^{16}O$ data cannot be reproduced by invoking a very low $\delta^{18}O_V$

value. With an extreme $\delta^{18}O_V$ of -30 ‰, the evaporation trajectory may be forced through those ponds below the envelope in $^{17}O$-excess, but that would result in an evaporation trajectory falling well above the majority of ponds in the diagram of d-excess over $\delta^{18}O$ (Fig. 11c-d). The model outcome is very well constrained in terms of variability in $\delta^{18}O_V$ because trends move in opposite directions between the two plots.

Sensitivity tests suggest a low impact of seasonal and diurnal variability in relative humidity, temperature, and wind conditions

on the isotopic composition of lakes and ponds in the Salar del Huasco (Fig. S4). To verify this, we compared recharge evaporation trajectories for daily mean (0-24h; h = 33 %, T = 4°C) and daytime mean conditions (6-18 h; h = 27 %, T = 7 %). Additionally, we modelled the recharge evaporation trajectory for daily mean relative humidity and temperature (h = 20 %, T = 10°C) weighted for the mean diurnal distribution of temperature and wind speed. High temperature and high wind speed amplify evaporation and were thus stronger weighted. Evaporation trajectories of all three scenarios fall close to each other in

triple oxygen isotope space (Fig. 11e), demonstrating that variability in relative humidity and temperature cannot account for the larger deviations of ponds from the recharge evaporation trajectory in the triple oxygen isotope plot that we attributed to either mixing or cut-off from recharge and evolution along a simple evaporation trajectory.

Collectively, these sensitivity analyses support our conclusion of mixing with flood water. This conclusion is further corroborated by the relationship between $\delta^{18}O$ and salinity of the ponds (Fig. S8). Most of the ponds show a significantly

higher salinity than expected from their degree of evaporation inferred from their isotopic composition. This may be partly caused by redissolution of previously precipitated salts. However, the fact that ponds particularly from the northern area fall on distinct lines indicates mixing of highly evaporated water with fresh flood water. Flooding in this case does not necessarily invoke inundation originating from the springs and inflow channels along the salar's margin, but may as well be the result of a rising groundwater table, which will mostly affect the more central, low-lying ponds that are also more evaporated. In fact,

all samples that indicate mixing, show advanced evaporation, i.e. plot rather to the right in the diagram. These results demonstrate the capability of the $^{17}O$-excess parameter to resolve mixing processes. In contrast, the d-excess parameter cannot distinguish mixing from evaporation at variable climatic boundary conditions at the Salar del Huasco (Fig. 11f).

**7 Conclusion**

When applied to the triple oxygen isotope system, the classic Craig-Gordon isotope evaporation model reliably predicts the

440 recharge evaporation trend of through-flow ponds within the hydrologically complex and seasonally dynamic groundwater-recharged Salar del Huasco, N-Chile, at ambient climatic boundary conditions and thus, by inference, from probably most lacustrine systems in general. The different hydrologic processes of evaporation and mixing of waters with different evaporation histories are demonstrably resolved in the triple oxygen isotope data. Cut-off from recharge during drought and

subsequent isotopic evolution on a simple pan evaporation trend is also observable in individual ponds. Likewise, flooding, i.e. the rapid mixing of evaporated water from the lake with fresh flood water of a different isotopic composition, may be identified using triple oxygen isotope data. In the classic $\delta^2H$–$\delta^{18}O$ system, the two types of evaporation trajectories as well as mixing curves overlap. Therefore, these processes are generally poorly resolvable with conventional $\delta^2H$–$\delta^{18}O$ measurements and the principle variable defining the hydrologic balance of a lake – i.e. the evaporation-to-inflow ratio (E/I) – cannot be determined unambiguously by the C-G model using the classic $\delta^2H$-$\delta^{18}O$ system alone.

A fundamental requirement for the application of the C-G model is an on-site estimate of the wind turbulence coefficient. Although the turbulence coefficient may affect the C-G model in a non-linear fashion at high wind speed (Gonfiantini et al., 2020), our on-site experiment suggests that a turbulence coefficient representing average ambient wind conditions may be estimated reasonably well by using the empirical relationship between d-excess and $\delta^{18}O$.

The C-G model – and thus, the critical E/I parameter – may thus be well constrained based on the analysis of all three isotope ratios – $^2H/^1H$, $^{17}O/^{16}O$ and $^{18}O/^{16}O$. – This is due to the fact that evaporation trajectories in plots of d-excess and $^{17}O$-excess over $\delta^{18}O$ evolve in opposite directions when input variables change. Triple oxygen isotope data are also capable of identifying mixing of precipitation with older, pre-evaporated connate water during infiltration in a groundwater recharge area. Combining the advantages of triple oxygen isotope data in terms of resolving fundamental hydrological processes – specifically mixing and the different types of evaporation with and without recharge – with the advantage of the classic $\delta^2H$-$\delta^{18}O$ system regarding the estimation of the critical wind turbulence exponent, a high level of accuracy in the hydrologic balance estimate of a lake – or an aquifer – may be achieved.

**Data availability**

All data reported herein is provided in the supplement and freely available at the Collaborative Research Centre 1211 database at https://doi.org/10.5880/CRC1211DB.36. It comprises the results of the ion concentration (ICP-OES) and isotopic analyses of water samples from ponds and lakes at the Salar del Huasco investigated in this study, as well as the isotopic data of the pan evaporation experiments carried out on-site.

**Author Contributions**

MS, CV and DH designed the study. CV, MS and CD were responsible for field work. CV conducted data analysis. CV, MS and DH evaluated the data. CV, MS and DH wrote the manuscript with contributions from CD.

**Competing Interests**

The authors declare that they have no conflict of interest.

**Acknowledgements**

This work was supported by the German Research Foundation (DFG) [268236062 – SFB 1211, subproject D03]. We thank Franc Megušar and Le-Tehnika (Kranj, Slovenia) for producing the Stirling cooler cycle system. Further, we thank the two anonymous reviewers whose constructive comments significantly improved the manuscript.

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

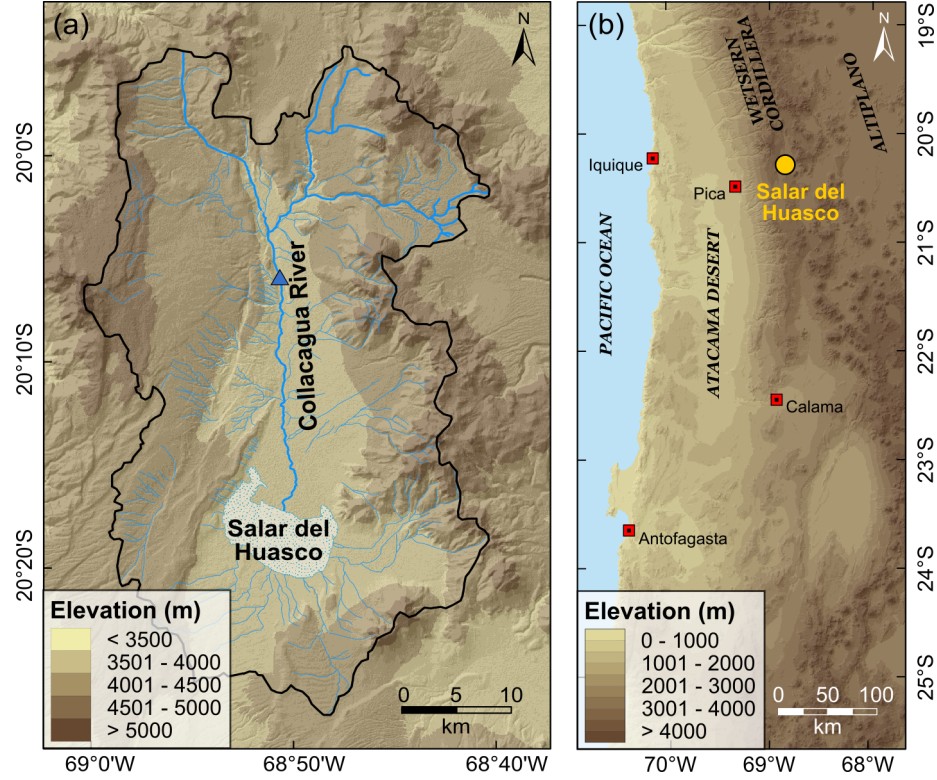


**Figure 1: Study Area. (a) Catchment of the Salar del Huasco (Salar del Huasco basin) with drainage. The blue triangle marks the sampling location of the Collacagua river. (b) Overview map. (DEM derived from SRTM data, created using ArcGIS 10.5.1)**

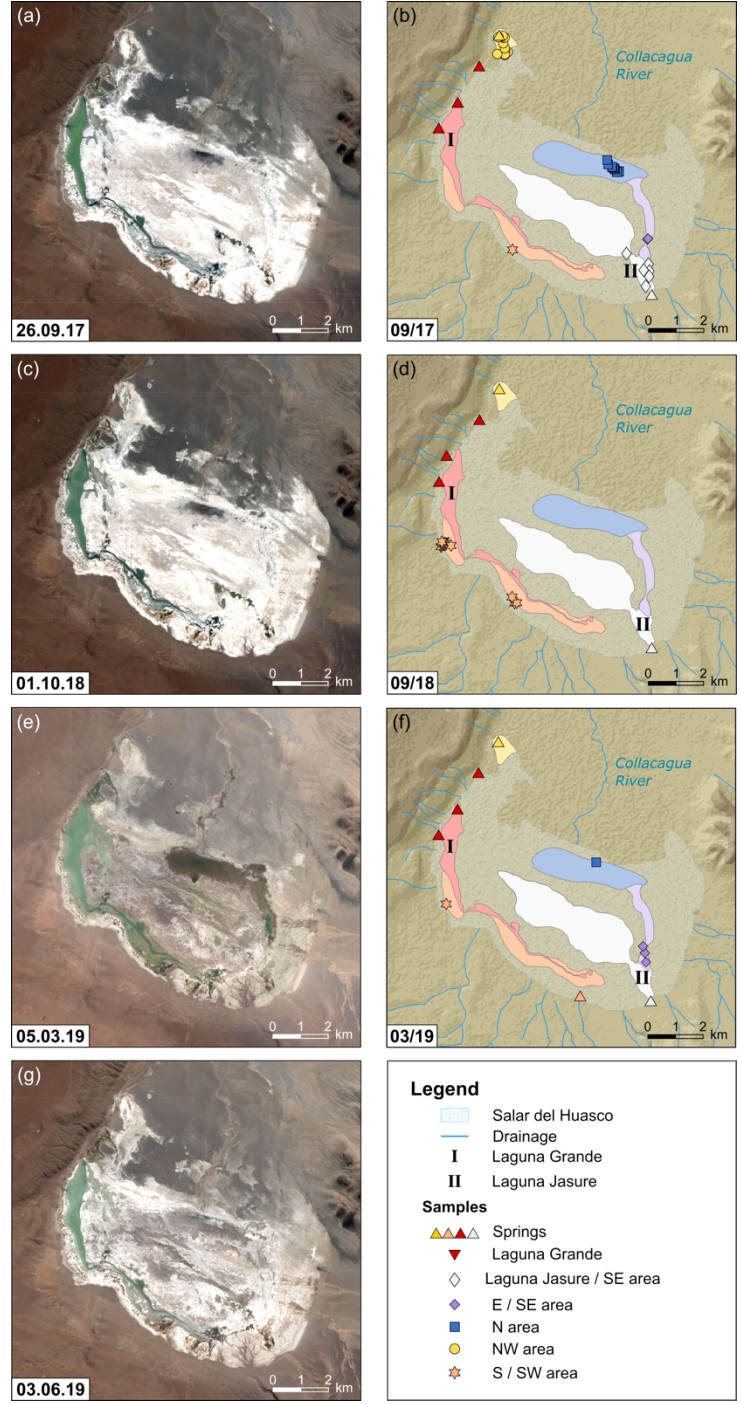

Figure 2: Illustration of the hydrological situation at the Salar del Huasco over the sampling period with sample locations for field campaigns in 09/17 ((a) and (b)), 09/18 ((c) and (d)), and 03/19 ((e) and (f)). Panel (g) reflects the hydrological situation at the Salar del Huasco 6 months after the last sampling campaign. Several hydrological subsystems (coloured areas) were identified based on satellite images (Copernicus Sentinel data, 2017, 2018, 2019) and field observations. Different symbols of sample locations refer to the corresponding hydrological subsystem (see text for details). (DEM derived from SRTM data, created using ArcGIS 10.5.1)


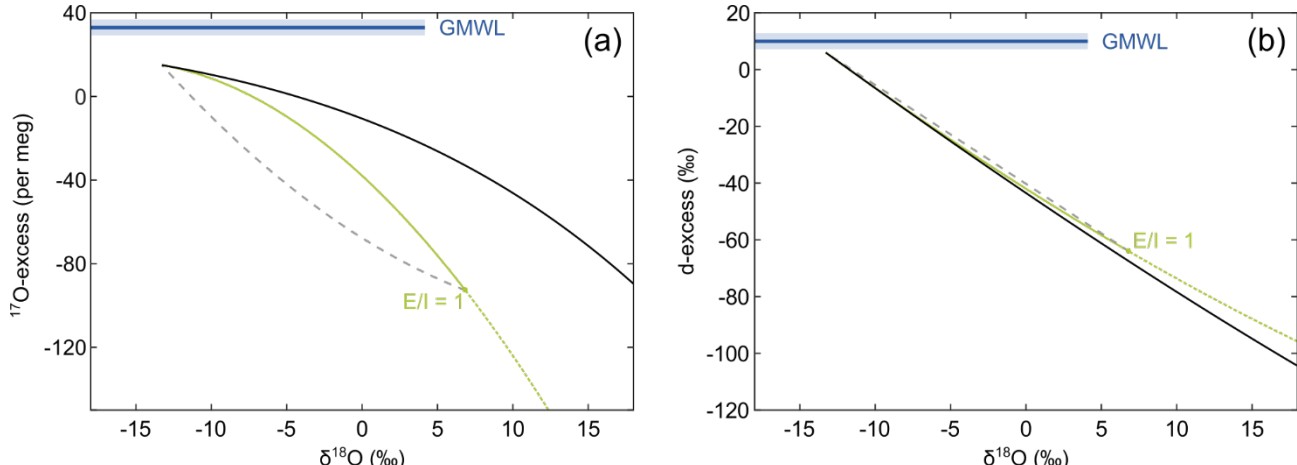

**Figure 3: Conceptual comparison of isotope effects associated with the three principal hydrological processes – simple evaporation without recharge, evaporation with recharge, and isotopic mixing – in the diagram of (a) $^{17}$O-excess over $\delta^{18}$O and (b) d-excess over $\delta^{18}$O. In the case of simple evaporation, the isotopic composition of an evaporating water body evolves along the black line.**
**Continuous recharge drives the isotopic composition of the evaporating water body below the simple evaporation trajectory onto the green line. The isotopic composition of a recharged lake is determined by the evaporation to inflow (E/I) ratio where increasing E/I lead to higher $\delta^{18}$O and lower $^{17}$O-excess values. In the case of a terminal lake (E/I = 1) all inflow is balanced by evaporation. The dashed grey line exemplifies admixture of fresh water similar in isotopic composition to the inflowing water, e.g. flood water, to the evaporated brine of a terminal lake.**

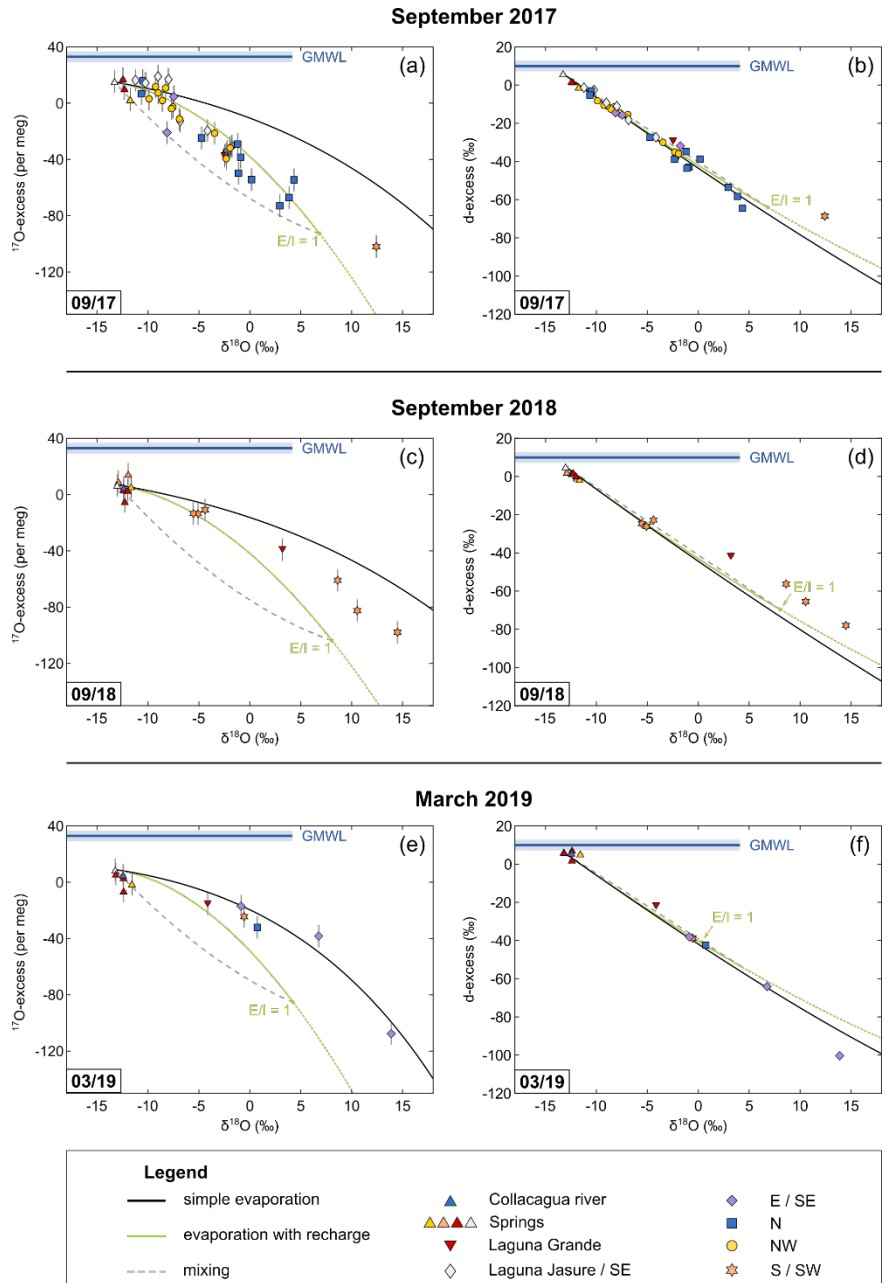


**Figure 4: Oxygen and hydrogen isotope data of the Collacagua river, springs, lakes and ponds sampled in the Salar del Huasco basin during field campaigns in 09/17 ((a) and (b)), 09/18 ((c) and (d)), and 03/19 ((e) and (f)). Colour coding refers to different hydrological subsystems (see legend and cf. Fig. 2). Note that the symbol size can be larger than the error bars. Trajectories for simple evaporation (black) and evaporation with recharge (green) were modelled using input parameters as summarized in Table 3. To calculate**
**temperature and relative humidity, daytime (6-18 h) conditions of the 10 (03/19) or 20 (09/17,09/18) prior to sampling were considered (cf. Sect. 6.3). Additionally, admixture of fresh groundwater (dashed grey line) as it might occur during flooding or snowmelt is exemplified for the case of a terminal lake (E/I = 1). The Global Meteoric Water Line (GMWL) serves as reference. Note that the apparent lack of samples on a recharge evaporation trajectory in 09/18 and 03/19 is simply the result of a sampling focus on sources and on ponds cut off from recharge during these campaigns.**


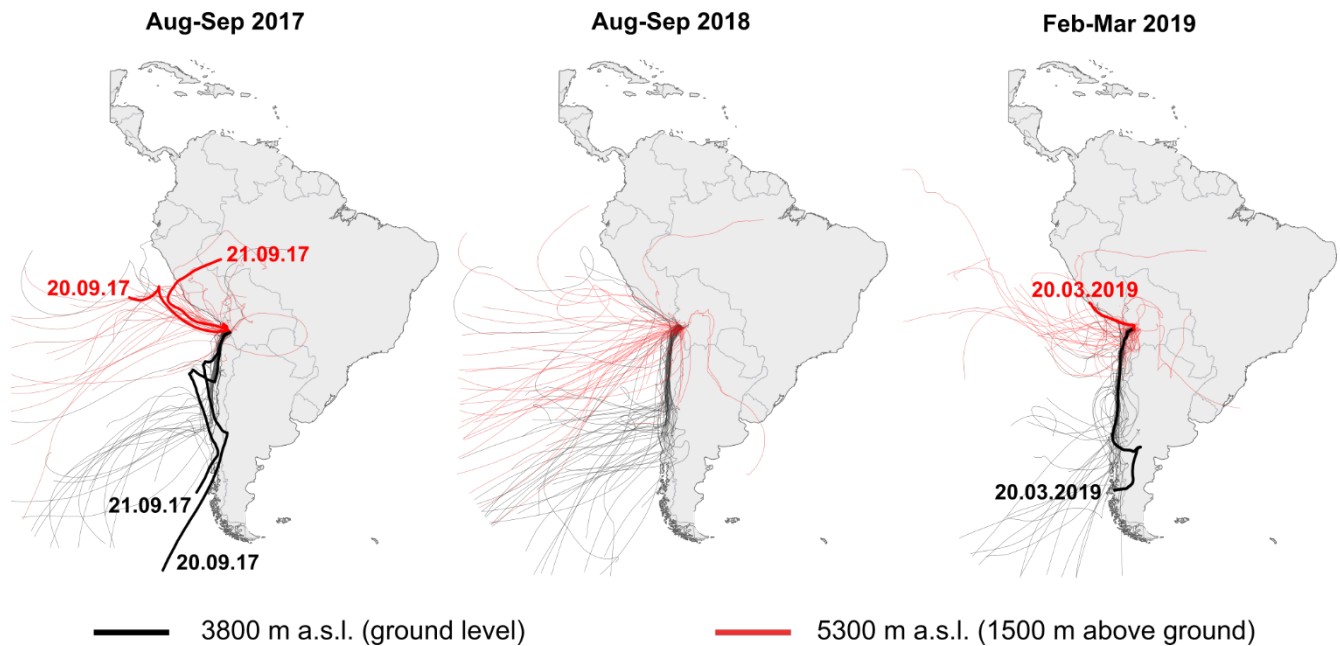

**Figure 5: HYSPLIT 7-day air mass back trajectories modelled for ground level at the Salar del Huasco (~3800 m above sea level (a.s.l.)) (black) and 1500 m above ground level (5300 m a.s.l.) (red) in daily resolution for the period from 23.08.2017 to 22.09.2017. The thick red and black trajectories represent modelled back trajectories for the time of vapour sampling on 20.09.17, 21.09.2017**

**and on 20.03.20**

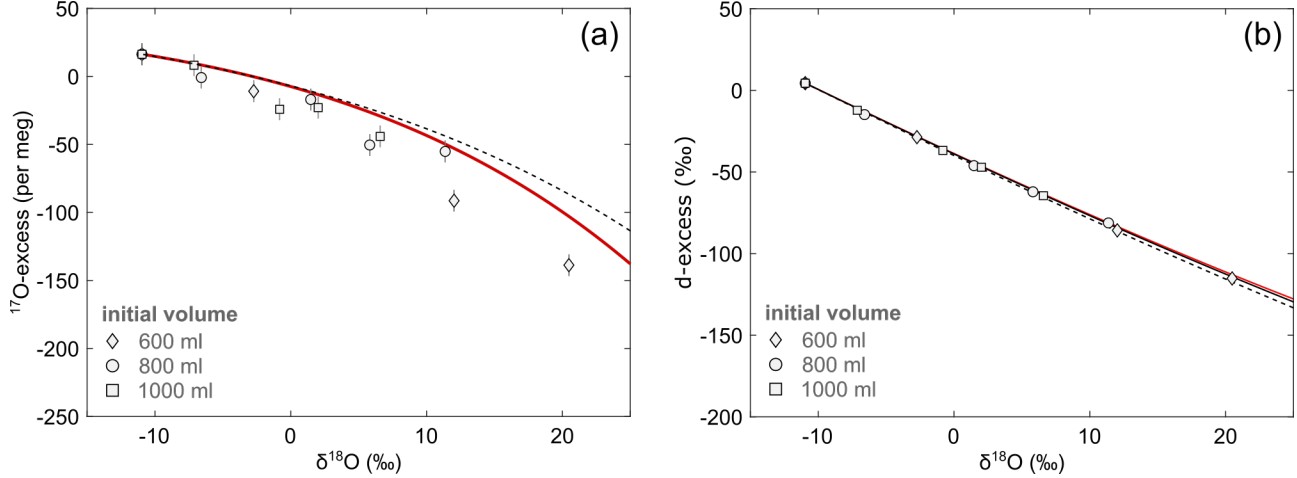

**Figure 6: Isotopic data of pan evaporation experiments in the diagrams of (a) $^{17}$O-excess over $\delta^{18}$O and (b) d-excess over $\delta^{18}$O. Pan evaporation experiments were carried out with different initial volume of 600 ml (diamonds), 800 ml (circles), 1000 ml (squares). Note that the symbol size can be larger than the error bars. The solid black line in panel (b) shows the best fitting evaporation trajectory (n = 0.55) for the simple evaporation model. The red lines represent the evaporation trajectories modelled using the virtual outlet model with n = 0.59 and an outlet of 20% (see text). Dashed lines illustrate the evaporation trajectories modelled with the simple evaporation model using the same turbulence coefficient as for the virtual outlet model. Model input parameters are summarized in Table 2.**

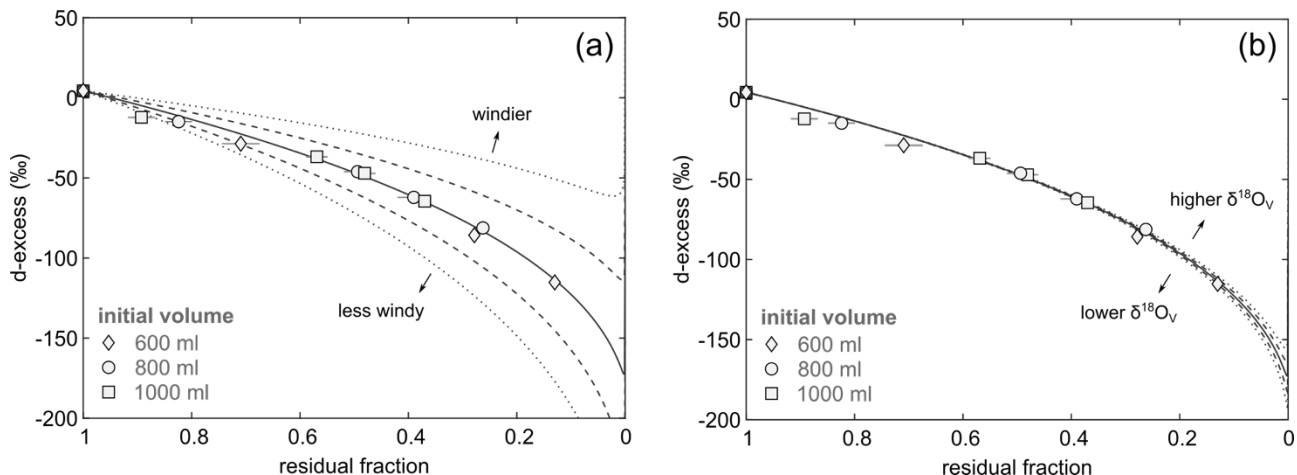

**Figure 7: Sensitivity of evaporation trajectories in a plot of d-excess over residual fraction to (a) the turbulence coefficient ($n$) and (b) the isotopic composition of atmospheric vapour ($\delta^{18}O_V$). Isotopic data of pan evaporation experiments with initial volume of 600 ml (diamonds), 800 ml (circles), 1000 ml (squares) are shown for comparison. Note that the symbol size is in most cases larger than the error bars. The solid line represents the modelled evaporation trajectory for simple evaporation using input parameters as summarized in Table 2 and a turbulence coefficient of 0.44. Dashed lines show model results for (a) different turbulence coefficients**
**in intervals of 0.1 and (b) varying $\delta^{18}O_V$ in intervals of 5 ‰, keeping other parameters constant.**

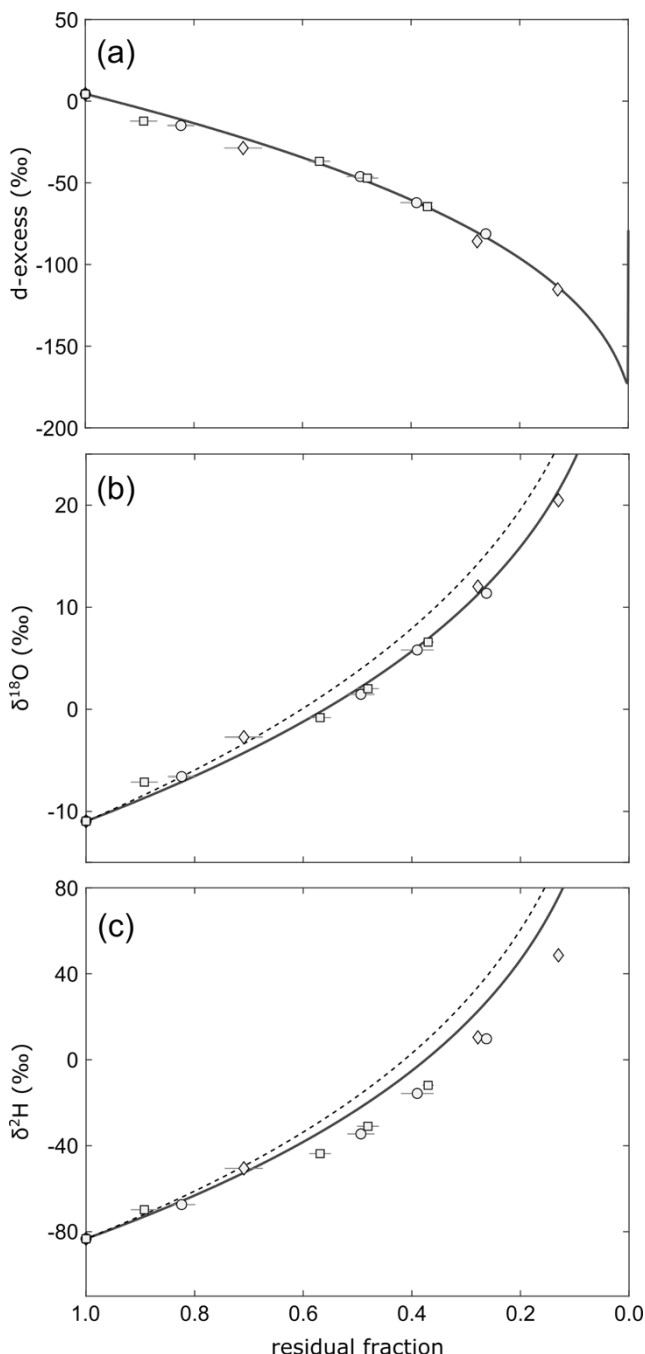

**Figure 8: Evolution of (a) d-excess, (b) $\delta^{18}O$, and (c) $\delta^2H$ of remaining water in the evaporation experiments. The solid lines show evaporation trajectories obtained by fitting the simple evaporation model independently to $\delta^{18}O$ (n= 0.34), $\delta^2H$ (n=0) and d-excess (n=0.44) vs residual fraction. Note that the $\delta^2H$ data cannot be fitted for any reasonable turbulence coefficient. The dashed lines in (a) and (b) are evaporation trajectories modelled using the same turbulence coefficient as for the d-excess data.**

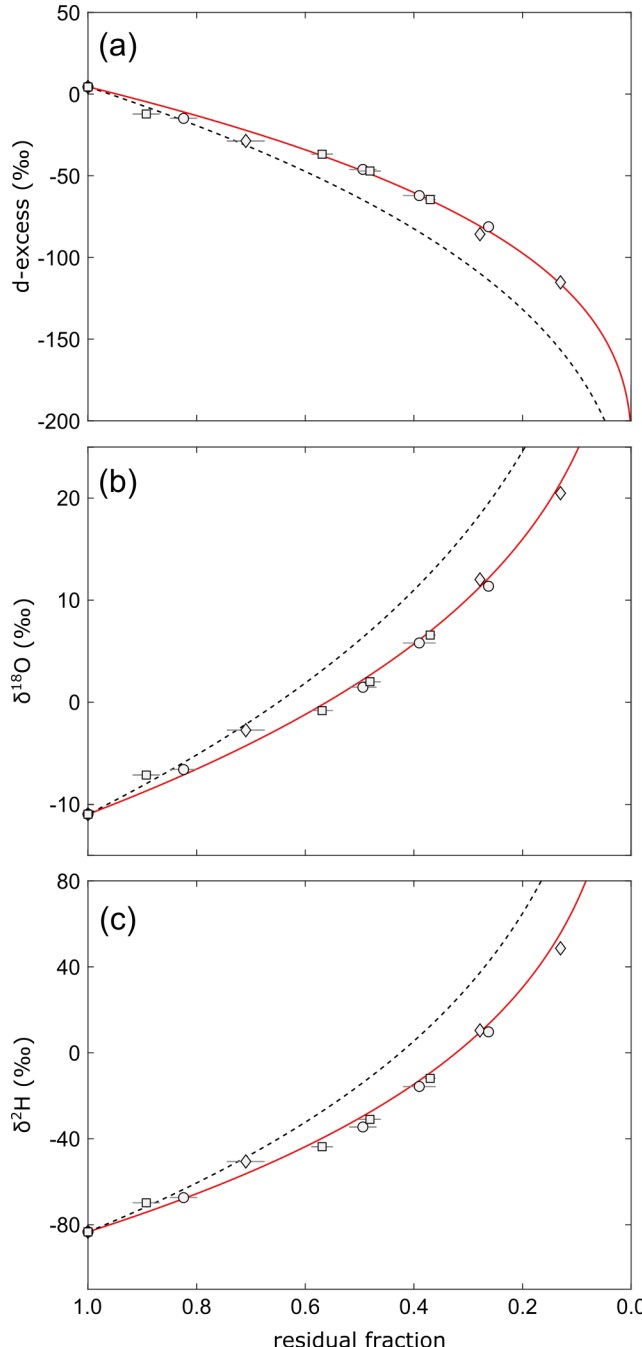

**Figure 9: Isotopic evolution of remaining water in evaporation experiments, similar to figure 8. Here, the red lines show evaporation trajectories obtained by fitting the virtual outlet model (n = 0.59, outlet = 20%). The dashed lines represent evaporation trajectories for the simple evaporation model obtained using the same turbulence coefficient as for the virtual outlet model.**

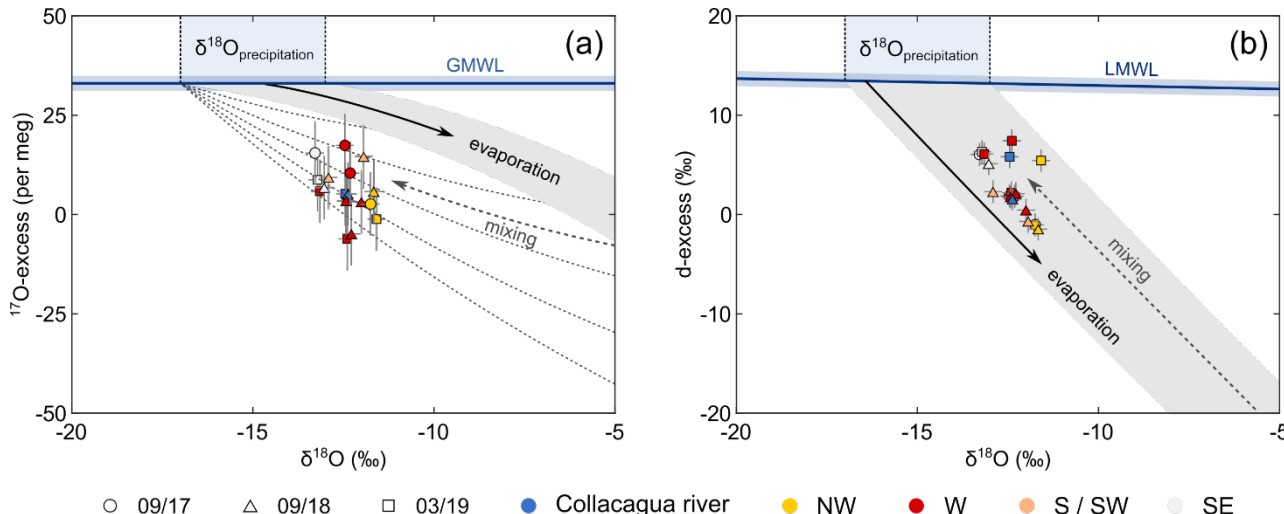


**Figure 10: Oxygen (a) and hydrogen (b) isotopic data of the Collacagua river and springs from the Salar del Huasco basin. Colour coding refers to different hydrological subsystems (cf. Fig. 2). The isotopic range of precipitation (-17 to -13 ‰) is derived from previously published data (Scheihing et al., 2018 and references therein). The local meteoric water line (LMWL: $\delta^2H=7.93\cdot\delta^{18}O+12.3$; Boschetti et al., 2019) was derived from precipitation data of northern Chile. Due to the lack of a comprehensive dataset of $^{17}O$ in precipitation, the global meteoric water line (GMWL) was used as reference in the triple oxygen isotope plot (a). The shaded grey area indicates the isotopic evolution of precipitation undergoing evaporation. The Collacagua river and all local springs fall below the common evaporation trend in a plot of $^{17}O$-excess over $\delta^{18}O$ (a). This offset can be explained considering mixing of precipitation with evaporated water in the vadose zone during infiltration into the soil. The effect of mixing is schematically illustrated by the dashed grey lines.**

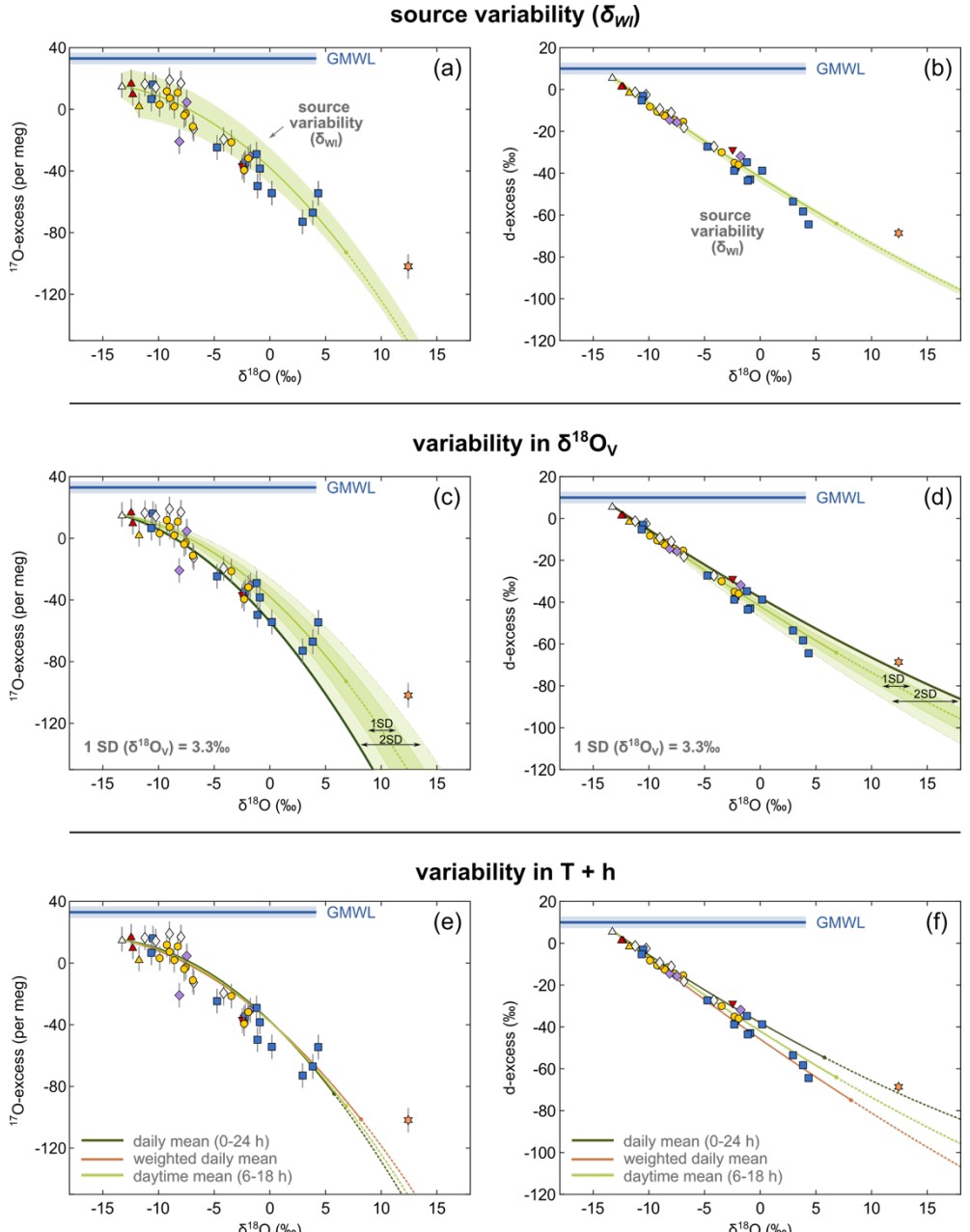

**Figure 11: Illustration of the impact of variability and uncertainty in the model input parameters on the evaporation trajectory exemplified for lakes and ponds sampled in 09/17.** The green line shows the modelled recharge evaporation trajectory for mean daytime conditions (6-18 h) at the Salar del Huasco. The shaded green areas in panel (a) and (b) illustrate the uncertainty introduced to the modelled trajectory by spatial variability in the isotopic composition of inflowing water ($\delta^{18}O_{WI}$). The dark solid lines represent corresponding evaporation trajectories in panel (a) and (b). Panel (c) and (d) show the impact of variable $\delta^{18}O_V$ on the evaporation trajectory illustrated by multiples intervals of the SD (1SD = 2.6 ‰; shaded areas). Panel (e) and (f) illustrate the effect of uncertainty in the assumed effective evaporation time interval. Recharge evaporation trajectories have been modelled for (1) daily mean (0-24 h; h = 33 %, T = 4°C) (dark green curve), (2) daily mean h = 20 % and T = 10°C weighted for the average diurnal distribution of temperature and wind speed (brown curve), and (3) daytime mean (6-18 h; h = 27 %, T = 7°C) conditions (light green curve).

**Table 1: Sampling intervals and isotopic data of atmospheric vapour.**

| Day | Time | $\delta^{17}O$ (‰) | $\delta^{18}O$ (‰) | $^{17}O$-excess (per meg) | n | $\delta^2H$ (‰) | d-excess (‰) | n |
|---|---|---|---|---|---|---|---|---|
| 20.09.2017 | 18:30-21:30 | -11.1 | -21.0 | 19 | 2 | -136 | 32 | 1 |
| 21.09.2017 | 18:30-21:30 | -9.4 | -17.8 | 16 | 2 | -114 | 28 | 1 |
| **mean** | | **-10.3** | **-19.4** | **18** | | **-125** | **30** | |
| **SD** | | **1.2** | **2.3** | **2.1** | | **15** | **2.8** | |
| 20.03.2019 | 10:00-13:00 | -12.9 | -24.3 | 33 | 2 | -154 | 41 | 1 |
| **overall average** | | **-11.1** | **-21.0** | **23** | | **-135** | **34** | |
| **SD** | | **1.7** | **3.2** | **9** | | **20** | **6** | |

**Table 2: Model input parameters of the simple evaporation model for pan evaporation experiments. The isotopic composition of initial water and atmospheric vapour was determined by measurement. The effective evaporation time interval was assumed to correspond to the period when T > 0°C to account for the freezing of water in the pans during the night.**

| Parameter | Value |
|---|---|
| $\delta^{18}O_{WI}$ (‰) | -11.0 |
| d-excess$_{WI}$ (‰) | 4 |
| $^{17}O$-excess$_{WI}$ (per meg) | 16 |
| $\delta^{18}O_V$ (‰) | -19.4 |
| d-excess$_V$ (‰) | 30 |
| $^{17}O$-excess$_V$ (per meg) | 18 |
| Temperature (°C) | 10 |
| Relative humidity (%) | 22 |

**Table 3: Model input parameters of the simple and the recharge evaporation model for natural ponds and lakes from the Salar del Huasco. The isotopic composition of inflowing water ($\delta_{WI}$) was inferred from the spring with the most depleted $\delta^{18}O$ value. The isotopic composition of atmospheric vapour ($\delta_V$) was estimated by direct measurement. Evaporation trajectories were modelled with temperature and relative humidity values averaged over 10 (03/19) or 20 days (09/17, 09/18) prior to sampling (Fig. 4), corresponding to the mean seasonal isotopic 'turnover-time' (c.f. Sect. 6.1). Diurnal variations in the evaporation rate were accounted for using daytime (6-18 h) values. The value of the turbulence coefficient was derived from pan evaporation experiment data (cf. Sect. 5.3).**

| Parameter | 2017 | 2018 | 2019 |
|---|---|---|---|
| $\delta^{18}O_{WI}$ (‰) | -13.3 | -13.0 | -13.2 |
| d-excess$_{WI}$ (‰) | 6 | 5 | 6 |
| $^{17}O$-excess$_{WI}$ (per meg) | 15 | 7 | 9 |
| $\delta^{18}O_V$ (‰) | -21.0 | -21.0 | -21.0 |
| d-excess$_V$ (‰) | 34 | 34 | 34 |
| $^{17}O$-excess$_V$ (per meg) | 23 | 23 | 23 |
| Temperature (°C) | 7 | 6 | 12 |
| Relative humidity (%) | 27 | 23 | 36 |
| Turbulence coefficient | 0.55 | 0.55 | 0.55 |