# Peer review of "Triple oxygen isotope systematics of evaporation and mixing processes in a dynamic desert lake system"

_Hydrology and Earth System Sciences, 2020_

## Referee Comment (RC1) · Anonymous Referee #1 · 20 Jul 2020

General Comments

The manuscript "Triple oxygen isotope systematics of evaporation and mixing processes in a dynamic desert lake system" explores the isotopic dynamics of a terminal lake system in Chile using evaporation pan experiments and Craig-Gordon evaporation modelling. The authors collected samples from numerous small lakes and ponds, groundwater springs, and atmospheric vapour to evaluate the effect of evaporation, sensitivity of input variables for the Craig-Gordon model, and assess the mixing of ponds. The study shows the highly sensitive nature of O17 in ponded water during fractionation and is an assess to partitioning water into mixed and evaporated water

pools.

I would recommend major revisions to improve the presentation of the manuscript. Three key issues need to be resolved as follows.

Firstly, the objectives and significance of the study are not clearly presented in the introduction. There is a limited introduction to the implication of using oxygen-17 other than "a potentially powerful tool" with much of the remaining introduction on oxygen-17 more suited to a methods section than an introduction. The importance of desert lake systems is central to this manuscript but is has limited emphasis only to oxygen-17. The objectives of the manuscript appeared to be only a sensitivity test of input variables in the Craig-Gordon model rather than assessing the dynamics of the salar system as a whole and using the Craig-Gordon model as a tool. The last part of the introduction seemed to be more of an abstract than an introduction and needs revision.

Secondly, the issues with the presentation of the methods and sampling are closely related to the third issue (results and discussion). Some of the information in "Sampling" belongs in "Study Site" (e.g. connectivity of ponds) and the section would benefit from more emphasis on the different conditions of each area during the sampling periods. Through the "Sampling", "Methods", and "Craig-Gordon" sections (as well as some introduction parts) there are terms that are not introduced properly or defined (e.g. d-excess, E/I). The "Sampling" section does not include the measurement height of the atmospheric data that was collected (temperature, relative humidity, $\delta v$), which may be significant for use in the Craig-Gordon model. The section on Craig-Gordon modelling lacked sufficient detail to allow for the replicability of the results. The formulation of the Craig-Gordon model used for oxygen-17 was not provided (I assume it is a similar form to Surma et al., 2018) which would be useful for the readers to understand the sensitivity assessed by the authors. It would also be useful if the authors would provide the other values used in the Craig-Gordon model (e.g $17\alpha l$-v_evap, $17\alpha l$-v_diff). Additionally, there is no information on how the authors accounted for mixing. Is it changes to the input end-member? Is it changes to the E/I ratio?

[Figure]

Thirdly, there are three main issues with the results and discussion section, the number of new methods introduced in the results, the amount of significance placed on few data points (vapour compositions), and the limited discussion of the results. Methods introduced in the results section include the HYSPLIT model (results shown without any previous mention of the model), translation of $\delta18Op$ (from OIPC) to $\delta18Ov$, Monte Carlo simulations and fitting of Craig-Gordon to evaporation pan data, and the set-up of sensitivity testing and the evaluation of the sensitivity. These components should all be introduced and described in the methods section. Through the results and discussion section, a lot of weight was placed on the atmospheric vapour compositions which were sampled over two days. While these samples are very important to constrain the Craig-Gordon model and an uncertrainty approach has been taken to assess some of the variability, the likelihood of large annual variability and impact should be discussed in more detail rather than discrediting the OPIC on two sample days. The discussion of the results is limited, particularly with the model uncertainty and the explanation of the dynamics of the salar, in context to the literature. Some ideas that may help the discussion could include (1) the impact of ice and high temperatures on evaporation pans and isotopic fractionation (2) the larger implications of model uncertainty, and (3) discussion on the causes of intra-annual changes of specific ponds (e.g causes of shifts in d-excess- or 17O-excess-$\delta18O$ space in Figs 9 & 10).

If the authors can make substantial improvements to clarifying the objectives and larger significance, describing methods, and expanding the discussion, the results could be a significant contribution for publication. Many of the issues above are described further in the Specific Comments section.

Specific Comments

P1L14: What is "recharge evaporation"

P1L19: Is it really a main finding to give a specific value of wind turbulence?

P1L21-26: What are the results that this tools give us? No need for all of the mineral

examples.

P1P26: Change the word "predicts" to a more relevant term

P1L27: Define $\alpha17$ and $\alpha18$

P1L27-29: Sentence not clear

P2L31: Define $\delta17O$ and $\delta18O$

P2L33: Define what "x" is

P2L36: Describe what 17O-excess shows (i.e. more negative is more fractionation/evaporation)

P2L47-58: The section is more of an abstract than an introduction/objectives

P3L72: How many years were used for long-term averages?

P3L81: Is there water loss from the lakes back to the groundwater system during low groundwater levels?

P3L86: Change the word "probably"

P4L98: Suggested change to "…18:00, but on the third day at 13:00".

P4L100: Where are the weather station and evaporation pan? What are the '+' markers in Figure S1? Is that water temperature?

P4L100: take out "at the experiment"

P4L104: Is the evaporation pan completely thawed at 9:30?

P4L114: Remove "i.e. the general direction of the Pacific coast".

P5L135: There was no previous mention of chemistry data. This data would be a useful discussion point in the manuscript for water sources and would help justify input sources for the Craig-Gordon model and the overall mixing of salar

P5L141: d-excess is not defined.

P5L141: Why is d-excess reported here when d-excess is dependent on $\delta$18O and $\delta$2H

P6L154: Change the wording of "classic evaporation theory".

P6L154: There are more than two scenarios; this manuscript is only examining two.

P6L156: "e.g. as a result of flooding or snowmelt". Meaning the addition of other source waters?

P6L157: What trajectories?

P6L158: What was used to create Figure 3? There isn't much discussion of what is on Figure 3.

P6L164: "initial or inflowing water". Should this be "initial and inflowing water"? In general, these two are different components. An assumption needs to be stated here that they are the same.

P6L167: Oceans have a value of $\approx$ 0.5, stating a theoretical value doesn't make sense here under natural conditions.

P6L170: Remove "classic"

P6L175: This is more results using data from the region rather than a method. Each subplot needs to be described or this should be in the supplementary materials.

P6L180: What about the diffusive properties? They could have a large effect on the results.

P6L182: Give the method used in Section 6.2 for determining n here

P6L182: What was the height of the measurement? Is it still used if it may not be representative? Not clear what value was used in the end.

[Figure]

P7L186-191: This should be in the study site section.

P7L194: This should be in methods. How many samples were taken? What was the temporal resolution?

P7L195-199: This should be introduced in the methods section if it is significant enough for a figure in the manuscript. Otherwise, the discussion can refer to it in the supplementary material (with a description of the model in the supplementary material).

P7L200: There needs to be a definition for the OIPC.

P7L200: How did you get the $\delta$18Ov from the precipitation? What values did you use for the correlation of precipitation to vapour? Also, the Bowen et al., 2005 reference should be for monthly values not annual values. Bowen et al., 2003 should be for annual

P7L204: "rainfall data", suggest a change to "rainfall isotopic estimates"

P7L205-208: Provide references that would suggest that the OIPC would not provide a reasonable annual or seasonal value of precipitation. The OIPC isn't intended for use on temporal scales less than one month (in an average year), so it would not be surprising if two samples may deviate from the average of a month.

P7L213: change the wording of "well constrainable"

P7L214: Suggest changing "...derived empirically from a plot..." to "...estimated from a best fit curve..."

P7L215: Fig 6a

P7L216: What is meant by "barely sensitive"?

P8L217: "this approach" suggest a change to "the C-G model"

P8L218: Why a monte carlo approach? There is only one unknown. There is no previous description of monte carlo simulation approach to the C-G model

P8L218: With the monte carlo approach did you take the best value? Is there no uncertainty with the Monte Carlo approach here?

P8L220-221: What about changes to evaporation due to the overheating of the evaporation pans?

P8L225: What about the effect of fractionation due to ice-freezing and thawing or through sublimation?

P8L230: Is the fitting done via a step-wise approach? Needs to be clarified.

P8L230: How much is "considerable sensitivity"?

P8L234: Is stating the $\delta$18Ov necessary?

P8L235: The description of obtaining $\delta$18Ov from the OIPC needs to be earlier

P8L239: Re-word the sentence. Why would it be tentative?

P9L251: Re-word the sentence. "fortunately"? The abstract and methods suggest that this value is well constrained. If it is not sufficiently well constrained then there should be a suggestion for further analysis and measurements.

P9L262: What about the LMWL?

P9L263: It could fall below the GMWL due to precipitation sources. It would be more relevant if this was compared to the LMWL

P9L265: Fig 8b not 7b

P9L266: Again Fig 8 not 7

P9L267: Again Fig 8 not 7

P9L271: Show the sample location on the site plot

P9L276-278: Where is the data to support the evaporation theory? What are the tributary values? How much enrichment is observed from the tributaries to the Collacagua

River?

P10L283: Evaporation and groundwater recharge are the only two aspects tested here, so should be the dominant factors. There needs to be a statement on how equal these factors are.

P10L290: "following". Following what?

P10L293-294: "two-spot measurement". Two measurements? Two-days of measurements? It is not clear how many measurements there are from the methods section.

P10L295: Suggest "estimated" rather than "derived"

P10L295: "Previously shown" shown in this study? Or which studies also show this?

P10L305: E/I was never introduced. Most figures include this as E/I = 0. Is figure 10 not E/I = 1? Where is the trajectory where E/I ≠ 1?

P10 L309: where does the assumption of ± 5 ‰ come from? How was this value determined? Is it from the uncertainty of the OIPC? Is it the range of monthly precipitation isotopes?

P11L317: Should be "oC" rather than "%"

P11L328: Which ponds are E/I > 0.5?

Section 6.4.2: Suggested that the causes of intra-annual changes (e.g. Figure 10) are discussed for different ponds. E.g. changes in E/I for a given year?

P12L349: As with the abstract, I would suggest clarifying "recharge evaporation" here. It is defined in the manuscript as an evaporation trajectory of a pond sourced by recharge that has evaporation, but it is not clear unless one line in the manuscript is read.

Figure 1: I would suggest adding in the measurement location of the Collacagua River here

Figure 2: For this figure and other figures, while the color scheme is good, I would suggest that the symbols be unified for the ponds (e.g. using square for all ponds/lakes and triangles for springs). The upside-down triangle (Laguna Grande) is difficult to identify on some plots).

Figure 3: What values were used to create the conceptual figure?

Figure 4: Is there an expectation of significant evaporation when the temperature is 0oC?

Figure 5: Is this figure necessary for the manuscript. There is a similar figure in the supplementary materials that would suffice. What are the thin lines on the figure?

Figure 6: What is the starting value of each interval?

Figure 9: Relabel the figure to make clearer. It is not clear that the left hand side shows the 17O-excess v. $\delta$18O while the right hand side it d-excess v. $\delta$18O

---

## Referee Comment (RC2) · Anonymous Referee #2 · 8 Sep 2020

The paper "Triple oxygen isotope systematics of evaporation and mixing processes in a dynamic desert lake system" by Voigt et al submitted for potential publication in HESS presents a field evaporation experiment with additional monitoring and sampling from the Huasco salt lake in the Chilean Altiplano desert for a triple oxygen isotope study and modelling with the Craig and Gordon model. The data shown is novel and there are not many publications in hydrology available that use oxygen-17 additionally to the much more common oxygen-18 and deuterium based stable water isotope studies. The study was also carried out in an extreme hydroclimatic environment providing new insights into the recharge and mixing processes of the salar. The novelty in hydrological process understanding could indeed be inferred from the oxygen-17 tracer additionally

to the other measured tracers and be supported by the modelling. I, therefore, see potential for this paper to contribute to the body of literature on isotope hydrology.

Having said that, I think that the paper could benefit from a thorough revision of structural aspects to clarify the key messages and conclusions. The paper does mix methods and results in many parts which I think rather confuses the reader. For example, already in the introduction you use a lot of detailed methods including equations 1 and 2 followed by your own results without stating clear research objectives. This might be a bit of a style questions and I see merit in this approach for a theoretical paper, but your paper is based on experimental work in a specific environment and falls out of the former category of scientific works. Therefore, I suggest to more generally introduce the potential utility and challenges of the oxygen-17 tracer in hydrology as this is still not widely used. You could also point towards the fact that you used IRMS and not a laser instrument. I would also urge the authors to present two or three specific objectives for clarity that can be used to guide the reader through the paper. In the methods, I found that the HYSPLIT analysis, the OIPC and the E/I modelling is not explained. I would also suggest to present the Craig and Gordon model with equation and in more detail in the methods clearly stating which parameters you varied to assess potential model uncertainty, how exactly you derived the wind turbulence parameter (this appears in the results) and the model experiments you are undertaking to assess the influence of measured atmospheric vapour isotope composition in the model. This leads me to suggest separating the results from the discussion and to only use two to three sub-headers that refer back to your specific results rather than at the moment 6 results sub-headers for clarity. These could be grouped according to field experiments, hydrological processes and model experiments as an example.

For the above reasons, I feel that this paper has potential but is not quite ready for publication in HESS and I invite the authors to consider my comments before the manuscript can be published with a clear message of the novel contributions.

Specific comments:

[Figure]

Title: I would suggest to substitute the term "systematics" with e.g. "dynamics" as systematics implies a general classification scheme of processes and their inter-relation, which I think is an overstatement for a case study.

Abstract: - I am not sure what you are referring to with the fundamental hydrologic process of recharge evaporation. If this is a new term you are defining it needs a proper definition and comes a bit early in the abstract. - I don't think you resolve the hydrologic balance in terms of a water balance of the lake as you don't calculate any fluxes.

Keywords are missing?

Table 2: Atmospheric vapour isotope composition was measured and not estimated.

---

## Author Comment (AC1) · 6 Oct 2020

**Reply to Referee #1**

We appreciate the constructive comments and suggestions of Reviewer #1. The provided comments have contributed substantially to improving the paper. Please, find below in black the comments of the reviewer, in blue our responses to the comments and how these comments will be addressed in the revised manuscript.

General Comments:

The manuscript "Triple oxygen isotope systematics of evaporation and mixing processes in a dynamic desert lake system" explores the isotopic dynamics of a terminal lake system in Chile using evaporation pan experiments and Craig-Gordon evaporation modelling. The authors collected samples from numerous small lakes and ponds, groundwater springs, and atmospheric vapour to evaluate the effect of evaporation, sensitivity of input variables for the Craig-Gordon model, and assess the mixing of ponds. The study shows the highly sensitive nature of O17 in ponded water during fractionation and is an assess to partitioning water into mixed and evaporated water pools. I would recommend major revisions to improve the presentation of the manuscript. Three key issues need to be resolved as follows.

Firstly, the objectives and significance of the study are not clearly presented in the introduction. There is a limited introduction to the implication of using oxygen-17 other than "a potentially powerful tool" with much of the remaining introduction on oxygen-17 more suited to a methods section than an introduction. The importance of desert lake systems is central to this manuscript but is has limited emphasis only to oxygen-17.  The objectives of the manuscript appeared to be only a sensitivity test of input variables in the Craig-Gordon model rather than assessing the dynamics of the salar system as a whole and using the Craig-Gordon model as a tool. The last part of the introduction seemed to be more of an abstract than an introduction and needs revision.

Thank you for pointing out that the objective of this study becomes not clear in the introduction. We thoroughly revised the introduction and focused on the larger implication on the 17O-excess parameter. Furthermore, we now better point out the main objectives of the manuscript to: 1) test the potential of triple oxygen isotope analyses to resolve fundamental hydrologic processes of evaporation and mixing of sources that cannot be resolved by the classical $\delta^2$H-$\delta^{18}$O analyses; 2) test the robustness of the Craig-Gordon model in a highly dynamic environment with considerable seasonal variability in all the model input parameters; and 3) demonstrate the potential of triple oxygen isotope analyses to derive the hydrological balance of lakes from water isotope and climate monitoring. The site of the Salar del Huasco was chosen because of its known extreme seasonality in order to obtain a maximum range of isotopic variability as a result of the above processes. The study's purpose was not to primarily investigate the seasonal dynamics of the Salar del Huasco in detail.

Secondly, the issues with the presentation of the methods and sampling are closely related to the third issue (results and discussion). Some of the information in "Sampling" belongs in "Study Site" (e.g. connectivity of ponds) and the section would benefit from more emphasis on the different conditions of each area during the sampling periods.

We have realized that there has been a doubling of information in the sections of "Study Area" (L81-94) and "Sampling" (L115-129). We complemented the paragraph in "Sampling" with the information given in "Study Area" and removed L81-94.

Through the "Sampling", "Methods", and "Craig-Gordon" sections (as well as some introduction parts) there are terms that are not introduced properly or defined (e.g. d-excess, E/I).

All variables in the main text are now defined properly. Furthermore, we added a detailed section on the principal isotope systematics in the supplement, in which all the formulas and variables that were used, are defined, and provided in a table, together with respective references.

The "Sampling" section does not include the measurement height of the atmospheric data that was collected (temperature, relative humidity, $\delta v$), which may be significant for use in the Craig-Gordon model.

We added this information.

The section on Craig-Gordon modelling lacked sufficient detail to allow for the replicability of the results. The formulation of the Craig-Gordon model used for oxygen-17 was not provided (I assume it is a similar form to Surma et al., 2018) which would be useful for the readers to understand the sensitivity assessed by the authors. It would also be useful if the authors would provide the other values used in the Craig-Gordon model (e.g $17\alpha l$-$v\_evap$, $17\alpha l$-$v\_diff$).

Indeed, we had not described the theoretical background of the isotope systematics and the Craig-Gordon model. We added a respective paragraph in the Craig-Gordon section. Details on the fractionation factors that were used are now given in the theoretical section in the supplement.

Additionally, there is no information on how the authors accounted for mixing. Is it changes to the input endmember? Is it changes to the E/I ratio?

There are various forms of mixing. In our case mixing may occurs episodically due to seasonal fluctuations in the groundwater table. When the groundwater table rises in the rainy season isotopically light groundwater is admixed to the evaporated ponds. We account for mixing by simply calculating two component-mass balance. We clarified this in the respective paragraph in the C-G section and added the mass balance equation.

Thirdly, there are three main issues with the results and discussion section, the number of new methods introduced in the results, the amount of significance placed on few data points (vapour compositions), and the limited discussion of the results. Methods introduced in the results section include the HYSPLIT model (results shown without any previous mention of the model), translation of $\delta18Op$ (from OIPC) to $\delta18Ov$, Monte Carlo simulations and fitting of Craig-Gordon to evaporation pan data, and the set-up of sensitivity testing and the evaluation of the sensitivity. These components should all be introduced and described in the methods section.

We have realized that the original manuscript had structural issues. We will follow these suggestions in the revised version of the manuscript as we have realised that it significantly improves the readability of the text and makes key messages clearer.

We added subsections on the determination of the turbulence coefficient, atmospheric vapour and the model sensitivity tests in the methods section. In these subsections, we mainly integrated information that was previously distributed in the study area or results/discussion sections. In the atmospheric vapour section, we now explained why the OIPC model is not suitable at our study site.

Through the results and discussion section, a lot of weight was placed on the atmospheric vapour compositions which were sampled over two days. While these samples are very important to constrain the Craig-Gordon model and an uncertainty approach has been taken to assess some of the variability, the likelihood of large annual variability and impact should be discussed in more detail rather than discrediting the OPIC on two sample days.

We want to make clear that we do not discredit the OIPC model. However, it is highly unlikely that the global OIPC model precipitation database (Bowen et al., 2020) accurately predicts vapour isotopic composition in regions when precipitation events are extremely rare. At the Salar del Huasco, precipitation occurs only seasonally and is generally derived from easterly sources, whereas westerly winds that do not produce precipitation, prevail during most times of the year (Aravena et al., 1999; Garreaud and Aceituno, 2001; Garreaud et al., 2003). We tested if vapour values at the salar are consistent with the prediction from the OIPC model.

We have measured an additional vapor sample that was taken during the field campaign in 03/2019 to strengthen the constraint of the atmospheric vapour value. The dataset remains small, but as pointed out by the reviewer, these data are very important. We also back up the data by verifying the measured vapor composition indirectly from the evaporation experiments. Fitting the C-G model through all experimental data using the measured $\delta^{18}O_V$ value results in a turbulence coefficient of n = 0.5 ± 0.15, which is in good agreement with the global range of reported values.

The discussion of the results is limited, particularly with the model uncertainty and the explanation of the dynamics of the salar, in context to the literature. Some ideas that may help the discussion could include (1) the impact of ice and high temperatures on evaporation pans and isotopic fractionation (2) the larger implications of model uncertainty, and(3) discussion on the causes of intra-annual changes of specific ponds (e.g. causes of shifts in d-excess- or 17O-excess-δ18O space in Figs 9 & 10).

This study focusses on the evaluation of the potential of the triple oxygen isotope system to distinguish hydrological processes of evaporation, recharge and mixing. Resolving intra-annual changes of individual ponds was beyond the scope of this study and requires a more detailed study of the different hydrological subsystems and monitoring at higher resolution. However, we discuss the effect of ice and variable temperatures on the evaporation pans as well as model uncertainty.

If the authors can make substantial improvements to clarifying the objectives and larger significance, describing methods, and expanding the discussion, the results could be a significant contribution for publication. Many of the issues above are described further in the Specific Comments section.

Specific Comments:

Some of the specific comments refer to issues already addressed in the general comments. We will implement changes as stated in the general comments above.

Minor comments regarding typos, rephrasing of sentences or adding of additional information on the sample location and the performance of sampling are all addressed in the revised version of the manuscript.

In the following, we will focus on specific questions that Reviewer #1 raised.

Specific comments related to the general comments:

P1L21-26: What are the results that these tools give us? No need for all of the mineral examples.

P1P26: Change the word "predicts" to a more relevant term

P1L27-29: Sentence not clear

P2L47-58: The section is more of an abstract than an introduction/objectives

The introduction was thoroughly revised.

P1L27: Define α17 and α18

P2L31: Define $\delta^{17}O$ and $\delta^{18}O$

P2L33: Define what "x" is

P2L36: Describe what $^{17}O$-excess shows (i.e. more negative is more fractionation/evaporation)

P5L141: d-excess is not defined.

We have added a detailed section on the principal isotope systematics in the supplement, in which all the formulas and variables that were used, are defined properly.

P6L182: Give the method used in Section 6.2 for determining n here

P7L186-191: This should be in the study site section.

P7L194: This should be in methods. How many samples were taken? What was the temporal resolution?

P7L195-199: This should be introduced in the methods section if it is significant enough for a figure in the manuscript. Otherwise, the discussion can refer to it in the supplementary material (with a description of the model in the supplementary material).

P8L235: The description of obtaining $\delta^{18}O_v$ from the OIPC needs to be earlier

These comments refer to restructuring of the manuscript and the addition of subsections in the "Methods". We have thoroughly revised the structure of the manuscript. We added subsections on atmospheric vapor/OIPC/HYSPLIT and the determination of the turbulence coefficient in the "Methods".

P7L200: There needs to be a definition for the OIPC.

P7L200: How did you get the $\delta^{18}O_v$ from the precipitation? What values did you use for the correlation of precipitation to vapour? Also, the Bowen et al., 2005 reference should be for monthly values not annual values. Bowen et al., 2003 should be for annual

P10L293-294: "two-spot measurement". Two measurements? Two-days of measurements? It is not clear how many measurements there are from the methods section.

P7L205-208: Provide references that would suggest that the OIPC would not provide a reasonable annual or seasonal value of precipitation. The OIPC isn't intended for use on temporal scales less than one month (in an average year), so it would not be surprising if two samples may deviate from the average of a month.

These comments address the issue of estimating the isotopic composition of vapour from the OIPC model or direct measurements. As stated in the 'General Comments', we do not want to discredit the OIPC model but we had to test its reliability for our very specific sample site.

P8L218: Why a monte carlo approach? There is only one unknown. There is no previous description of monte carlo simulation approach to the C-G model

P8L218: With the monte carlo approach did you take the best value? Is there no uncertainty with the Monte Carlo approach here?

P8L230: Is the fitting done via a step-wise approach? Needs to be clarified.

P8L230: How much is "considerable sensitivity"?

These comments refer to the determination of the turbulence coefficient. We added a subsection describing the method how to determine the turbulence coefficient in the methods section. We will provide errors for the determined turbulence coefficient estimated from the uncertainty of input parameters. For this purpose, sensitivity tests were performed, which will be provided in the supplement.

P10 L309: where does the assumption of ± 5 ‰ come from? How was this value determined? Is it from the uncertainty of the OIPC? Is it the range of monthly precipitation isotopes?

We will add details on how the uncertainty of individual model parameters was estimated and extend the discussion on model results in the revised manuscript.

P3L81: Is there water loss from the lakes back to the groundwater system during low groundwater levels?

We think, loss of water from the lake by "infiltration" additionally to evaporation at low groundwater levels is likely, but we do not have evidence for this. However, this would not affect the isotopic composition of the remaining pond water as infiltration should not lead to isotope fractionation.

P4L104: Is the evaporation pan completely thawed at 9:30?

No, it is not. But we do not know the exact time interval, which will also shift a bit depending on the amount of frozen water and the amount of remaining water. Because the exact time interval is unknown, we the period when T > 0°C as the effective time interval. Even if the time interval was delayed, this has only a minor effect on the average temperature and relative humidity values. Furthermore, sensitivity tests demonstrate that even high uncertainty in T and rH have no significant impact the modelled turbulence coefficient.

P5L135: There was no previous mention of chemistry data. This data would be a useful discussion point in the manuscript for water sources and would help justify input sources for the Craig-Gordon model and the overall mixing of salar.

We agree, but this topic is beyond the scope of this manuscript. However, we now mention that the TDS data confirm mixing in plots of $\delta^{18}O$ vs salinity.

P5L141: Why is d-excess reported here when d-excess is dependent on δ18O and δ2H

We are not sure to what this question is referred to. Meteoric waters fall on a trend line, where the slope is mainly defined by equilibrium fractionation during condensation. Kinetic processes, e.g. evaporation progress along shallower slopes. These processes cannot be distinguished by δ18O or δ2H alone. The d-excess parameter quantifies the offset from this slope and thus better visualizes the proportion of different fractionation effects in the $\delta^2H$-$\delta^{18}O$ system.

P6L156: "e.g. as a result of flooding or snowmelt". Meaning the addition of other source waters?

Meaning one event in which e.g. flooding water is mixed into the pond water. In such short-term events the isotopic composition of the pond water may be affected by a mixing process (simple mass balance between two water masses ['pristine' floodwater and pre-evaporated pond water]) rather than by a change in recharge conditions. We added a few sentences explaining how we account for these mixing events in the "Craig-Gordon" section (see above).

P6L158: What was used to create Figure 3? There isn't much discussion of what is on Figure 3.

It's a schematic figure showing that different trajectories for simple evaporation, recharge evaporation, and mixing can be well distinguished in triple oxygen isotope space (A), while they merge in data uncertainty in the plot of d-excess vs. $\delta^{18}O$ (B). The discrimination of these three trajectories is almost independent of the input variables (h, T, n, $\delta_{WI}$, $\delta_V$). The Craig-Gordon equations that were used to model the trajectories for simple and recharge evaporation were added. Furthermore, the paragraph on mixing was extended (see general comments).

P6L167: Oceans have a value of≈0.5, stating a theoretical value doesn't make sense here under natural conditions.

The data from Uemura et al. (2010) imply a turbulence coefficient n = 0.3 (due to sea spray….), but this value is likely biased due to sea spray. Other than that, we are not aware of any published n < 0.5, indicating that such conditions are at least rare.

P6L180: What about the diffusive properties? They could have a large effect on the results.

We state that h, T, $\delta_{WI}$ and $\delta_V$ can be obtained by direct measurement or monitoring, but the turbulence coefficient, which accounts for diffusivity conditions, is not easily obtainable (Line 180 – 182). We accurately quantify the turbulence coefficient from isotopic data of pan evaporation experiments.

P8L220-221: What about changes to evaporation due to the overheating of the evaporation pans?

Good point. Temperature and relative humidity in the C-G equation are related to the evaporating water surface rather than air. Air temperature and water temperature are often assumed to be equal, but we observed temperature differences between air and water of about ± 5°C. However, sensitivity tests with this temperature range show that uncertainty in the actual temperature value have no significant impact on the modelled turbulence coefficient.

P8L225: What about the effect of fractionation due to ice-freezing and thawing or through sublimation?

Indeed, isotope fractionation effects due to freezing biased our first estimate of the turbulence coefficient as discussed in detail in the text. We then use a second estimate that is not affected by ice-freezing.

P9L262: What about the LMWL?

P9L263: It could fall below the GMWL due to precipitation sources. It would be more relevant if this was compared to the LMWL

A large $\delta^2H$-$\delta^{18}O$ dataset of meteoric waters from the Altiplano region suggests that the LMWL in the $\delta^2H$-$\delta^{18}O$ system is similar to the GMWL (Boschetti et al. 2019 and references therein). Such a dataset does not exist for, triple oxygen isotopes, hence there is no determined LMWL for $^{17}O$-excess. In figure 8, we used the LMWL in the plot of d-excess vs $\delta^{18}O$, but the GMWL in the plot of $^{17}O$-excess vs $\delta^{18}O$.

P9L276-278: Where is the data to support the evaporation theory? What are the tributary values? How much enrichment is observed from the tributaries to the Collacagua River?

We have only measured the isotopic composition of the Collacagua river and cannot constrain the isotopic composition of its tributaries by own measurements. However, Uribe et al. (2015) investigated the isotopic composition of rivers in the Huasco basin. They demonstrated that the isotopic composition of surface water reflects the isotopic composition of precipitation in the source region, which varies with altitude (more depletion in 18O with increasing altitude). "Downstream the waters become enriched (…), because they receive contributions of water recharged at lower altitudes, which are characterized by a more enriched isotopic content than the water coming from higher altitudes, and also due to the effect of evaporation along the riverbed." (Uribe et al., 2015). We will rewrite this paragraph in the revised version of the manuscript and refer to this previous study.

P10L283: Evaporation and groundwater recharge are the only two aspects tested here, so should be the dominant factors. There needs to be a statement on how equal these factors are.

P10L305: E/I was never introduced. Most figures include this as E/I = 0. Is figure 10 not E/I = 1? Where is the trajectory where E/I6= 1?

The recharge evaporation trajectory is modelled in dependence of the E/I ratio. The equations are now provided in the Craig-Gordon section. The position of the measured isotopic composition of ponds on the trajectory gives an indication of the throughflow rate (close to source water = high recharge, close to terminal lake = low recharge, below terminal lake = evaporation exceeds inflow and the lake tends to desiccate).

Section 6.4.2: Suggested that the causes of intra-annual changes (e.g. Figure 10) are discussed for different ponds. E.g. changes in E/I for a given year?

Monitoring at higher resolution and more information on recharge/evaporation rates as well as about the subsurface groundwater regime are required to quantify intra-annual changes in E/I in individual ponds.

Minor comments:

P1L19: Is it really a main finding to give a specific value of wind turbulence?

P1L14: What is "recharge evaporation"

P3L72: How many years were used for long-term averages?

P3L86: Change the word "probably"

P4L98: Suggested change to "…18:00, but on the third day at 13:00".

P4L100: Where are the weather station and evaporation pan? What are the '+' markers in Figure S1? Is that water temperature?

P4L100: take out "at the experiment"

P4L114: Remove "i.e. the general direction of the Pacific coast".

P6L154: Change the wording of "classic evaporation theory".

P6L154: There are more than two scenarios; this manuscript is only examining two.

P6L157: What trajectories?

P6L164: "initial or inflowing water". Should this be "initial and inflowing water"? In general, these two are different components. An assumption needs to be stated here that they are the same.

P6L170: Remove "classic"

P6L175: This is more results using data from the region rather than a method. Each subplot needs to be described or this should be in the supplementary materials.

P6L182: What was the height of the measurement? Is it still used if it may not be representative? Not clear what value was used in the end.

P7L204: "rainfall data", suggest a change to "rainfall isotopic estimates"

P7L213: change the wording of "well constrainable"

P7L214: Suggest changing "...derived empirically from a plot..." to "...estimated from a best fit curve..."

P7L215: Fig 6a

P7L216: What is meant by "barely sensitive"?

P8L217: "this approach" suggest a change to "the C-G model"

P8L234: Is stating the $\delta^{18}O_v$ necessary?

P8L239: Re-word the sentence. Why would it be tentative?

P8L251: Re-word the sentence. "fortunately"? The abstract and methods suggest that this value is well constrained. If it is not sufficiently well constrained, then there should be a suggestion for further analysis and measurements.

P9L265: Fig 8b not 7b

P9L266: Again Fig 8 not 7

P9L267: Again Fig 8 not 7

P9L271: Show the sample location on the site plot

P10L290: "following". Following what?

P10L295: Suggest "estimated" rather than "derived"

P10L295: "Previously shown" shown in this study? Or which studies also show this?

P11L317: Should be "°C" rather than "%"

P11L328: Which ponds are E/I > 0.5?

P12L349: As with the abstract, I would suggest clarifying "recharge evaporation" here. It is defined in the manuscript as an evaporation trajectory of a pond sourced by recharge that has evaporation, but it is not clear unless one line in the manuscript is read.

There are a number of minor comments regarding typos, rephrasing of sentences or simple adding of information on sample location or height of measurements. We will follow these suggestions in the revised version of our manuscript.

Comments on figures:

Figure 1: I would suggest adding in the measurement location of the Collacagua River here

Figure 2: For this figure and other figures, while the colour scheme is good, I would suggest that the symbols be unified for the ponds (e.g. using square for all ponds/lakes and triangles for springs). The upside-down triangle (Laguna Grande) is difficult to identify on some plots).

Figure 3: What values were used to create the conceptual figure?

Figure 4: Is there an expectation of significant evaporation when the temperature is 0°C?

Figure 5: Is this figure necessary for the manuscript. There is a similar figure in the supplementary materials that would suffice. What are the thin lines on the figure?

Figure 6: What is the starting value of each interval?

Figure9: Relabel the figure to make clearer. It is not clear that the left-hand side shows the $^{17}$O-excess v. $\delta^{18}$O while the right hand side it d-excess v. $\delta^{18}$O.

We are grateful for the detailed review and these useful suggestions that help to improve the illustration of our data. We will implement these suggestions in the revised version of the manuscript in terms of what we consider as a meaningful improvement of the presentation of our data. We will carefully evaluate which figures should be shown in the main text and which should be included in the supplement.

*References:*

*Aravena, R., Suzuki, O., Peña, H., Pollastri, A., Fuenzalida, H., Grilli, A., 1999. Isotopic composition and origin of the precipitation in Northern Chile. Appl. Geochemistry 14, 411–422. https://doi.org/10.1016/S0883-2927(98)00067-5*

*Boschetti, T., Cifuentes, J., Iacumin, P., Selmo, E., 2019. Local meteoric water line of northern Chile (18°S-30°S): An application of error-in-variables regression to the oxygen and hydrogen stable isotope ratio of precipitation. Water 11, 1–16. https://doi.org/10.3390/w11040791*

Bowen, G. J. (202) The Online Isotopes in Precipitation Calculator, version 3.1. http://www.waterisotopes.org.

Garreaud, R., Vuille, M., Clement, A.C., 2003. The climate of the Altiplano: Observed current conditions and mechanisms of past changes. Palaeogeogr. Palaeoclimatol. Palaeoecol. 194, 5–22. https://doi.org/10.1016/S0031-0182(03)00269-4

Garreaud, R.D., Aceituno, P., 2001. Interannual Rainfall Variability over the South American Altiplano. J. Clim. 14, 2779–2789. https://doi.org/10.1175/1520-0442(2001)014<2779:IRVOTS>2.0.CO;2

Uemura, R., Barkan, E., Abe, O., Luz, B., 2010. Triple isotope composition of oxygen in atmospheric water vapor. Geophys. Res. Lett. 37, 1–4. https://doi.org/10.1029/2009GL041960

Uribe, J., Muñoz, J.F., Gironás, J., Oyarzún, R., Aguirre, E., Aravena, R., 2015. Assessing groundwater recharge in an Andean closed basin using isotopic characterization and a rainfall-runoff model: Salar del Huasco basin, Chile. Hydrogeol. J. 23, 1535–1551. https://doi.org/10.1007/s10040-016-1383-1

---

## Author Comment (AC2) · 6 Oct 2020

**Reply to Referee #2**

We appreciate the helpful comments and suggestions of Reviewer #2. The provided comments mainly concern structural aspects, which were also criticised by Reviewer #1. We will follow these suggestions in the revised version of the manuscript as we have realised that it significantly improves the readability of the text and makes key messages clearer. Please, find below in black the comments of the reviewer, in blue our responses to the comments and how the comments will be addressed in the revised manuscript.

General Comments:

The paper "Triple oxygen isotope systematics of evaporation and mixing processes in a dynamic desert lake system" by Voigt et al submitted for potential publication in HESS presents a field evaporation experiment with additional monitoring and sampling from the Huasco salt lake in the Chilean Altiplano desert for a triple oxygen isotope study and modelling with the Craig and Gordon model. The data shown is novel and there are not many publications in hydrology available that use oxygen-17 additionally to the much more common oxygen-18 and deuterium based stable water isotope studies. The study was also carried out in an extreme hydroclimatic environment providing new insights into the recharge and mixing processes of the salar. The novelty in hydrological process understanding could indeed be inferred from the oxygen-17 tracer additionally to the other measured tracers and be supported by the modelling. I, therefore, see potential for this paper to contribute to the body of literature on isotope hydrology.

Having said that, I think that the paper could benefit from a thorough revision of structural aspects to clarify the key messages and conclusions. The paper does mix methods and results in many parts which I think rather confuses the reader.

For example, already in the introduction you use a lot of detailed methods including equations 1 and 2 followed by your own results without stating clear research objectives. This might be a bit of a style questions and I see merit in this approach for a theoretical paper, but your paper is based on experimental work in a specific environment and falls out of the former category of scientific works. Therefore, I suggest to more generally introduce the potential utility and challenges of the oxygen-17 tracer in hydrology as this is still not widely used. You could also point towards the fact that you used IRMS and not a laser instrument. I would also urge the authors to present two or three specific objectives for clarity that can be used to guide the reader through the paper.

This is a point that was also considered by reviewer #1. We thoroughly revised the introduction and focused on the larger implication on the 17O-excess parameter with a minimum of equations that are necessary to understand how 17O-excess and d-excess are derived. A detailed section regarding the terminology will be provided in the supplement. Furthermore, we now better point out the main objectives of the manuscript to: 1) test the potential of triple oxygen isotope analyses to resolve fundamental hydrologic processes of evaporation and mixing of sources that cannot be resolved by the classical $\delta^2$H-$\delta^{18}$O analyses; 2) test the robustness of the Craig-Gordon model in a highly dynamic environment with considerable seasonal variability in all the model input parameters; and 3) demonstrate the potential of triple oxygen isotope analyses to derive the hydrological balance of lakes from water isotope and climate monitoring. The site of the Salar del Huasco was chosen because of its known extreme seasonality in order to obtain a maximum range of isotopic variability as a result of the above processes. The study's purpose was not to primarily investigate the seasonal dynamics of the Salar del Huasco in detail.

In the methods, I found that the HYSPLIT analysis, the OIPC and the E/I modelling is not explained. I would also suggest to present the Craig and Gordon model with equation and in more detail in the methods clearly stating which parameters you varied to assess potential model uncertainty, how exactly you derived the wind turbulence parameter (this appears in the results) and the model experiments you are undertaking to assess the influence of measured atmospheric vapour isotope composition in the model.

In the revised manuscript, we have added subsections on the determination of the turbulence coefficient, atmospheric vapour and the model sensitivity tests in the methods sections. We have also added a paragraph describing the Craig-Gordon model with the major equations. Details on terminology and the Craig-Gordon model will be provided in the supplement.

This leads me to suggest separating the results from the discussion and to only use two to three sub-headers that refer back to your specific results rather than at the moment 6 results sub-headers for clarity. These could be grouped according to field experiments, hydrological processes and model experiments as an example.

We revised the structure of the results and discussion section but we found that separating results and discussion rigidly would in this specific case hinder the flow of reading.

For the above reasons, I feel that this paper has potential but is not quite ready for publication in HESS and I invite the authors to consider my comments before the manuscript can be published with a clear message of the novel contributions.

We have now updated the manuscript considerably, following the reviewer's suggestions. It is now better suitable for a broader audience with clear messages on the novelty of the $^{17}$O-excess parameter to investigate evaporation and mixing.

Specific Comments:

Title: I would suggest to substitute the term "systematics" with e.g. "dynamics" as systematics implies a general classification scheme of processes and their inter-relation, which I think is an overstatement for a case study.

The term dynamics is already included in the title. It is unclear why the triple oxygen isotope systematics for our case study should not apply globally. Mixing and evaporation occur in all such environments.

Abstract: - I am not sure what you are referring to with the fundamental hydrologic process of recharge evaporation. If this is a new term you are defining it needs a proper definition and comes a bit early in the abstract. - I don't think you resolve the hydrologic balance in terms of a water balance of the lake as you don't calculate any fluxes.

Recharge evaporation is now explained. It refers to a pond that is constantly recharged while water is lost via evaporation and outflow.

The C-G model is designed to estimate steady-state (i.e. the water balance is constant). However, building a proper flux model requires a solid understanding of all relevant fluxes. The C-G model is a fundamental step in this direction. In fact, we already discuss fluxes that are important e.g. to understand how fast a pond adapts to steady state.

Keywords are missing?

Keywords are not requested at this point of submission, but will be provided.

Table 2: Atmospheric vapour isotope composition was measured and not estimated.

We will change this accordingly.

---

## Author Response (AR1)

Dear Mrs Ali

Thank you for inviting us to revise our manuscript entitled "Triple oxygen and isotope systematics of evaporation and mixing processes in a dynamic desert lake system". We are grateful to both you and the two reviewers for your detailed and constructive comments and suggestions. Please find attached a revised version of our contribution. A detailed point-by-point

5  response to the reviewer's comments was already provided in the individual author responses. Therefore, only the key issues unveiled by the reviewers are addressed below. A marked-up version of the revised manuscript is provided at the bottom of this file.

A major critical point concerned structural aspects. The revised version of the manuscript now follows a classical structure with separate Results and Discussion, which indeed improves the readability of the text, making key messages clearer. In

10  detail, we have revised the introduction section to better point out the objectives of the manuscript. We have added a paragraph describing the Craig-Gordon model with major equations. Details on terminology and a table summarizing all variables used within the manuscript are now provided in the supplement. As suggested by both reviewers, we added a subsection on atmospheric vapour in the Methods section, where we now make clear why the OIPC model may not be suitable at our study site.

15  In the revised manuscript, we now discuss the possible isotope effect introduced by the freezing and thawing of our pan evaporation experiment. Additionally, we now include the results of laboratory experiments recently published by Gonfiantini et al. (2020) and compared them with our experimental data. These new findings greatly improve our understanding of the turbulence coefficient.

We have added a subsection on the isotopic residence times in ponds and lakes at the Salar del Huasco and extended our

20  discussion on the model uncertainty to address the critique of one reviewer, who stated that we cannot infer information on the hydrological lake balance using isotope systematics. We show that combined triple oxygen and hydrogen isotope systematics do allow inferences on hydrological lake balance and fundamental hydrological processes.

Yours Sincerely

25  Claudia Voigt (on behalf of all co-authors)

35 **Point-to-point response**

Reply to Referee #1

The comments of reviewer #1 have contributed substantially to improving the paper. Please, find below in black the comments of the reviewer, in blue our responses to the comments and how these comments are addressed in the revised manuscript.

Firstly, the objectives and significance of the study are not clearly presented in the introduction. There is a limited introduction to the implication of using oxygen-17 other than "a potentially powerful tool" with much of the remaining introduction on oxygen-17 more suited to a methods section than an introduction. The importance of desert lake systems is central to this manuscript but is has limited emphasis only to oxygen-17. The objectives of the manuscript appeared to be only a sensitivity test of input variables in the Craig-Gordon model rather than assessing the dynamics of the salar system as a whole and using the Craig-Gordon model as a tool. The last part of the introduction seemed to be more of an abstract than an introduction and needs revision.

We thoroughly revised the introduction and focus now on hydrological processes that determine the isotopic composition of lakes and the larger implication of the $^{17}O$-excess parameter as a tool to resolve these processes as well as to identify changes in the hydrological balance of lakes. We now stress the main objectives of the manuscript, which are: 1) test the potential of triple oxygen isotope analyses to resolve fundamental hydrologic processes of evaporation and mixing of sources that cannot be resolved by the classical $\delta^2H$-$\delta^{18}O$ analyses; 2) test the robustness of the Craig-Gordon model in a highly dynamic environment with considerable seasonal variability in all the model input parameters; and 3) demonstrate the potential of triple oxygen isotope analyses to derive the hydrological balance of lakes from water isotope and climate monitoring.

Secondly, the issues with the presentation of the methods and sampling are closely related to the third issue (results and discussion). Some of the information in "Sampling" belongs in "Study Site" (e.g. connectivity of ponds) and the section would benefit from more emphasis on the different conditions of each area during the sampling periods. Through the "Sampling", "Methods", and "Craig-Gordon" sections (as well as some introduction parts) there are terms that are not introduced properly or defined (e.g. d-excess, E/I). The "Sampling" section does not include the measurement height of the atmospheric data that was collected (temperature, relative humidity, $\delta v$), which may be significant for use in the Craig-Gordon model. The section on Craig-Gordon modelling lacked sufficient detail to allow for the replicability of the results. The formulation of the Craig-Gordon model used for oxygen-17 was not provided (I assume it is a similar form to Surma et al., 2018) which would be useful for the readers to understand the sensitivity assessed by the authors. It would also be useful if the authors would provide the other values used in the Craig-Gordon model (e.g 17αl-v_evap, 17αl-v_diff). Additionally, there is no information on how the authors accounted for mixing. Is it changes to the input endmember? Is it changes to the E/I ratio?

We now address all of these points. In the revised manuscript we have now included a theoretical section with major equations regarding the isotope systematics, the Craig-Gordon model, and mixing. All variables are now defined in the main text and, additionally, a detailed section on terminology and a table summarizing all variables and fractionation factors used within the manuscript are provided in the supplement. Missing information in the sampling section were added and doubling of information given in the Study Area and the Sampling section were removed.

Thirdly, there are three main issues with the results and discussion section, the number of new methods introduced in the results, the amount of significance placed on few data points (vapour compositions), and the limited discussion of the results. Methods introduced in the results section include the HYSPLIT model (results shown without any previous mention of the model), translation of $\delta 18Op$ (from OIPC) to $\delta 18Ov$, Monte Carlo simulations and fitting of Craig-Gordon to evaporation pan data, and the set-up of sensitivity testing and the evaluation of the sensitivity. These components should all be introduced and described in the methods section. Through the results and discussion section, a lot of weight was placed on the atmospheric vapour compositions which were sampled over two days. While these samples are very important to constrain the Craig-Gordon model and an uncertainty approach has been taken to assess some of the variability, the likelihood of large annual variability and impact should be discussed in more detail rather than discrediting the OPIC on two sample days.

We added a subsection on atmospheric vapour in the methods section, where we now explained why the OIPC model is possibly not suitable for our study site and introduce the HYSPLIT modelling. The sections on the turbulence coefficient and the model sensitivity tests were thoroughly revised so that derived conclusions become clearer and more concise. This discussion should now also clarify why we had placed so much weight on the vapour composition in our previous version. It is a rather critical parameter for the C-G model.

It is certainly not our intention to discredit the OIPC model. The question we raise is if a ''precipitation model'' is applicable to a region where precipitation is virtually absent, which is now better explained. Our small dataset was expanded by one sample and we also back up the data by verifying the measured vapour composition indirectly from the evaporation experiments. As pointed out by the reviewer, constraints on vapour isotopic compositions are very important.

The discussion of the results is limited, particularly with the model uncertainty and the explanation of the dynamics of the salar, in context to the literature. Some ideas that may help the discussion could include (1) the impact of ice and high temperatures on evaporation pans and isotopic fractionation (2) the larger implications of model uncertainty, and(3) discussion on the causes of intra-annual changes of specific ponds (e.g. causes of shifts in d-excess- or 17O-excess-$\delta 18O$ space in Figs 9 & 10).

We now address these points. In the revised manuscript, we discuss the effect of ice and variable temperatures on the evaporation pans and the effect of sublimation (i.e. reviewers point 1). We have extended the evaluation of model uncertainty (i.e. reviewers point 2). Moreover, we have now included the results of laboratory experiments recently published by Gonfiantini et al. (2020) and compared them with our experimental data, which greatly improves our understanding of the turbulence coefficient. We added a subsection on the isotopic residence times in ponds and lakes at the Salar del Huasco to show that combined triple oxygen and hydrogen isotope systematics do allow inferences on hydrological lake balance and fundamental hydrological processes. Absolute constraints on the timescales of isotopic shifts as a function of pond depth are now provided (i.e. reviewers point 3).

We again appreciate the helpful comments and suggestions of Reviewer #2. We followed most of these suggestions in the revised version of the manuscript. Please, find below in black the comments of the reviewer, in blue how these are addressed in the revised manuscript.

Having said that, I think that the paper could benefit from a thorough revision of structural aspects to clarify the key messages and conclusions. The paper does mix methods and results in many parts which I think rather confuses the reader. For example, already in the introduction you use a lot of detailed methods including equations 1 and 2 followed by your own results without stating clear research objectives. This might be a bit of a style questions and I see merit in this approach for a theoretical paper, but your paper is based on experimental work in a specific environment and falls out of the former category of scientific works. Therefore, I suggest to more generally introduce the potential utility and challenges of the oxygen-17 tracer in hydrology as this is still not widely used. You could also point towards the fact that you used IRMS and not a laser instrument. I would also urge the authors to present two or three specific objectives for clarity that can be used to guide the reader through the paper.

The structure of the paper has been changed as suggested. We thoroughly revised the introduction and focus now on hydrological processes that determine the isotopic composition of lakes and the larger implication of the 17O-excess parameter as a tool to resolve these processes as well as to identify changes in the hydrological balance of lakes. We used a minimum of equations that are necessary to understand how 17O-excess and d-excess are derived. We now better point out the main objectives of the manuscript to: 1) test the potential of triple oxygen isotope analyses to resolve fundamental hydrologic processes of evaporation and mixing of sources that cannot be resolved by the classical $\delta 2H$-$\delta 18O$ analyses; 2) test the robustness of the Craig-Gordon model in a highly dynamic environment with considerable seasonal variability in all the model input parameters; and 3) demonstrate the potential of triple oxygen isotope analyses to derive the hydrological balance of lakes from water isotope and climate monitoring. The fact that we use IRMS and not a laser instrument is stated in the Methods section and in our opinion not relevant enough to be mentioned in the introduction.

In the methods, I found that the HYSPLIT analysis, the OIPC and the E/I modelling is not explained. I would also suggest to present the Craig and Gordon model with equation and in more detail in the methods clearly stating which parameters you varied to assess potential model uncertainty, how exactly you derived the wind turbulence parameter (this appears in the results) and the model experiments you are undertaking to assess the influence of measured atmospheric vapour isotope composition in the model.

In the revised manuscript we have now included a theoretical section with major equations regarding the isotope systematics, the Craig-Gordon model, and mixing. All variables are now defined in the main text and, additionally, a detailed section on terminology and a table summarizing all variables and fractionation factors used within the manuscript are provided in the supplement. Furthermore, we added a subsection on atmospheric vapour and introduce the HYSPLIT modelling. The OPIC model is widely known and explained in detail in the cited literature.

135 This leads me to suggest separating the results from the discussion and to only use two to three sub-headers that refer back to your specific results rather than at the moment 6 results sub-headers for clarity. These could be grouped according to field experiments, hydrological processes and model experiments as an example.

The revised version of the manuscript now follows a classical structure with separate Results and Discussion, which indeed improves the readability of the text, making key messages clearer. The results section is now divided in three subsections

140 focusing on natural waters in the Salar del Huasco, atmospheric vapour and the isotopic results of pan evaporation experiments. Subsections in the discussion provide now key messages of our data including the determination of the turbulence coefficient from pan evaporation experiment data, the potential of the triple oxygen isotope system to resolve multiple generations of infiltration in groundwater aquifers, and its power to resolve individual 
[revised manuscript text omitted]

| | | |
|---|---|---|
| **Seite 6: [1] hat gelöscht** | **Claudia Voigt** | **19.11.20 17:30:00** |
| **Seite 8: [2] hat gelöscht** | **Claudia Voigt** | **19.11.20 17:30:00** |
| **Seite 8: [3] hat gelöscht** | **Claudia Voigt** | **19.11.20 17:30:00** |
| **Seite 14: [4] hat gelöscht** | **Claudia Voigt** | **19.11.20 17:30:00** |
| **Seite 14: [5] hat gelöscht** | **Claudia Voigt** | **19.11.20 17:30:00** |
| **Seite 14: [6] hat gelöscht** | **Claudia Voigt** | **19.11.20 17:30:00** |
| **Seite 14: [7] hat gelöscht** | **Claudia Voigt** | **19.11.20 17:30:00** |
| **Seite 14: [8] hat gelöscht** | **Claudia Voigt** | **19.11.20 17:30:00** |
| **Seite 14: [9] hat gelöscht** | **Claudia Voigt** | **19.11.20 17:30:00** |
| **Seite 14: [10] hat gelöscht** | **Claudia Voigt** | **19.11.20 17:30:00** |
| **Seite 16: [11] hat gelöscht** | **Claudia Voigt** | **19.11.20 17:30:00** |
| **Seite 16: [12] hat gelöscht** | **Claudia Voigt** | **19.11.20 17:30:00** |
| **Seite 16: [13] hat gelöscht** | **Claudia Voigt** | **19.11.20 17:30:00** |
| **Seite 16: [14] hat gelöscht** | **Claudia Voigt** | **19.11.20 17:30:00** |
| **Seite 19: [15] hat gelöscht** | **Claudia Voigt** | **19.11.20 17:30:00** |

**Seite 23: [19] Formatiert**      **Claudia Voigt**      **19.11.20 17:30:00**

Einzug: Links: 0 cm, Hängend: 0.85 cm, Keine Absatzkontrolle, Leerraum zwischen asiatischem und westlichem Text nicht anpassen, Leerraum zwischen asiatischem Text und Zahlen nicht anpassen

**Seite 23: [20] Formatiert**      **Claudia Voigt**      **19.11.20 17:30:00**

Keine Absatzkontrolle, Leerraum zwischen asiatischem und westlichem Text nicht anpassen, Leerraum zwischen asiatischem Text und Zahlen nicht anpassen

**Seite 27: [21] hat gelöscht**      **Claudia Voigt**      **19.11.20 17:30:00**

**Seite 32: [22] hat gelöscht**      **Claudia Voigt**      **19.11.20 17:30:00**

**Seite 33: [23] hat gelöscht**      **Claudia Voigt**      **19.11.20 17:30:00**

**Seite 33: [23] hat gelöscht**      **Claudia Voigt**      **19.11.20 17:30:00**

**Seite 33: [23] hat gelöscht**      **Claudia Voigt**      **19.11.20 17:30:00**

**Seite 33: [23] hat gelöscht**      **Claudia Voigt**      **19.11.20 17:30:00**

**Seite 33: [23] hat gelöscht**      **Claudia Voigt**      **19.11.20 17:30:00**

**Seite 33: [23] hat gelöscht**      **Claudia Voigt**      **19.11.20 17:30:00**

**Seite 33: [23] hat gelöscht**      **Claudia Voigt**      **19.11.20 17:30:00**

**Seite 33: [23] hat gelöscht**      **Claudia Voigt**      **19.11.20 17:30:00**

**Seite 33: [23] hat gelöscht**      **Claudia Voigt**      **19.11.20 17:30:00**

**Seite 33: [23] hat gelöscht**      **Claudia Voigt**      **19.11.20 17:30:00**

| Seite 33: [23] hat gelöscht | Claudia Voigt | 19.11.20 17:30:00 |
|---|---|---|

| Seite 33: [23] hat gelöscht | Claudia Voigt | 19.11.20 17:30:00 |
|---|---|---|

| Seite 33: [23] hat gelöscht | Claudia Voigt | 19.11.20 17:30:00 |
|---|---|---|

| Seite 34: [24] hat gelöscht | Claudia Voigt | 19.11.20 17:30:00 |
|---|---|---|

| Seite 34: [25] hat gelöscht | Claudia Voigt | 19.11.20 17:30:00 |
|---|---|---|

| Seite 34: [26] hat formatiert | Claudia Voigt | 19.11.20 17:30:00 |
|---|---|---|

Schriftfarbe: Text 1

| Seite 34: [27] hat gelöscht | Claudia Voigt | 19.11.20 17:30:00 |
|---|---|---|

| Seite 34: [28] hat gelöscht | Claudia Voigt | 19.11.20 17:30:00 |
|---|---|---|

| Seite 34: [29] hat formatiert | Claudia Voigt | 19.11.20 17:30:00 |
|---|---|---|

Schriftart: 10 Pt.

| Seite 34: [30] Formatiert | Claudia Voigt | 19.11.20 17:30:00 |
|---|---|---|

Zentriert, Zeilenabstand:  Mehrere 1.15 ze

| Seite 34: [31] hat formatiert | Claudia Voigt | 19.11.20 17:30:00 |
|---|---|---|

Schriftart: 10 Pt.

| Seite 34: [32] Formatiert | Claudia Voigt | 19.11.20 17:30:00 |
|---|---|---|

Zentriert, Zeilenabstand:  Mehrere 1.15 ze

| Seite 34: [33] Formatiert | Claudia Voigt | 19.11.20 17:30:00 |
|---|---|---|

Zeilenabstand:  Mehrere 1.15 ze

| Seite 34: [34] hat formatiert | Claudia Voigt | 19.11.20 17:30:00 |
|---|---|---|

Schriftart: 10 Pt.

| Seite 34: [35] Formatiert | Claudia Voigt | 19.11.20 17:30:00 |
|---|---|---|

Zentriert, Zeilenabstand:  Mehrere 1.15 ze

| Seite 34: [36] Formatiert | Claudia Voigt | 19.11.20 17:30:00 |
|---|---|---|

Zeilenabstand:  Mehrere 1.15 ze

| Seite 34: [39] Formatiert | Claudia Voigt | 19.11.20 17:30:00 |
|---|---|---|

Zeilenabstand:  Mehrere 1.15 ze

| Seite 34: [40] hat formatiert | Claudia Voigt | 19.11.20 17:30:00 |
|---|---|---|

Schriftart: 10 Pt.

| Seite 34: [41] Formatiert | Claudia Voigt | 19.11.20 17:30:00 |
|---|---|---|

Zentriert, Zeilenabstand:  Mehrere 1.15 ze

| Seite 34: [42] Formatiert | Claudia Voigt | 19.11.20 17:30:00 |
|---|---|---|

Zeilenabstand:  Mehrere 1.15 ze

| Seite 34: [43] hat formatiert | Claudia Voigt | 19.11.20 17:30:00 |
|---|---|---|

Schriftart: 10 Pt.

| Seite 34: [44] Formatiert | Claudia Voigt | 19.11.20 17:30:00 |
|---|---|---|

Zentriert, Zeilenabstand:  Mehrere 1.15 ze

| Seite 34: [45] Formatiert | Claudia Voigt | 19.11.20 17:30:00 |
|---|---|---|

Zeilenabstand:  Mehrere 1.15 ze

| Seite 34: [46] hat formatiert | Claudia Voigt | 19.11.20 17:30:00 |
|---|---|---|

Schriftart: 10 Pt.

| Seite 34: [47] Formatiert | Claudia Voigt | 19.11.20 17:30:00 |
|---|---|---|

Zentriert, Zeilenabstand:  Mehrere 1.15 ze

| Seite 34: [48] Formatiert | Claudia Voigt | 19.11.20 17:30:00 |
|---|---|---|

Zeilenabstand:  Mehrere 1.15 ze

| Seite 34: [49] hat formatiert | Claudia Voigt | 19.11.20 17:30:00 |
|---|---|---|

Schriftart: 10 Pt.

| Seite 34: [50] Formatiert | Claudia Voigt | 19.11.20 17:30:00 |
|---|---|---|

Zentriert, Zeilenabstand:  Mehrere 1.15 ze

| Seite 34: [51] Formatiert | Claudia Voigt | 19.11.20 17:30:00 |
|---|---|---|

Zeilenabstand:  Mehrere 1.15 ze

| Seite 34: [52] hat formatiert | Claudia Voigt | 19.11.20 17:30:00 |
|---|---|---|

Schriftart: 10 Pt.

| Seite 34: [53] hat formatiert | Claudia Voigt | 19.11.20 17:30:00 |
|---|---|---|

Schriftart: 10 Pt.

| Seite 34: [54] Formatiert | Claudia Voigt | 19.11.20 17:30:00 |
|---|---|---|

Zentriert, Zeilenabstand:  Mehrere 1.15 ze

| Seite 34: [57] hat formatiert | Claudia Voigt | 19.11.20 17:30:00 |
|---|---|---|

Schriftart: 10 Pt.

| Seite 34: [58] hat gelöscht | Claudia Voigt | 19.11.20 17:30:00 |
|---|---|---|

| Seite 35: [59] hat formatiert | Claudia Voigt | 19.11.20 17:30:00 |
|---|---|---|

Schriftart: Kursiv

| Seite 35: [59] hat formatiert | Claudia Voigt | 19.11.20 17:30:00 |
|---|---|---|

Schriftart: Kursiv

| Seite 35: [60] hat gelöscht | Claudia Voigt | 19.11.20 17:30:00 |
|---|---|---|

| Seite 35: [60] hat gelöscht | Claudia Voigt | 19.11.20 17:30:00 |
|---|---|---|

| Seite 35: [60] hat gelöscht | Claudia Voigt | 19.11.20 17:30:00 |
|---|---|---|

| Seite 35: [60] hat gelöscht | Claudia Voigt | 19.11.20 17:30:00 |
|---|---|---|

| Seite 35: [60] hat gelöscht | Claudia Voigt | 19.11.20 17:30:00 |
|---|---|---|

| Seite 35: [61] hat formatiert | Claudia Voigt | 19.11.20 17:30:00 |
|---|---|---|

Schriftart: 10 Pt.

| Seite 35: [62] Formatiert | Claudia Voigt | 19.11.20 17:30:00 |
|---|---|---|

Zeilenabstand:  Mehrere 1.15 ze

| Seite 35: [63] Formatiert | Claudia Voigt | 19.11.20 17:30:00 |
|---|---|---|

Zentriert, Zeilenabstand:  Mehrere 1.15 ze

| Seite 35: [64] hat formatiert | Claudia Voigt | 19.11.20 17:30:00 |
|---|---|---|

Schriftart: 10 Pt.

| Seite 35: [65] Formatiert | Claudia Voigt | 19.11.20 17:30:00 |
|---|---|---|

Zeilenabstand:  Mehrere 1.15 ze

| Seite 35: [66] hat formatiert | Claudia Voigt | 19.11.20 17:30:00 |
|---|---|---|

Schriftart: 10 Pt.

| Seite 35: [67] Formatiert | Claudia Voigt | 19.11.20 17:30:00 |
|---|---|---|

Zeilenabstand:  Mehrere 1.15 ze

| Seite 35: [70] Formatiert | Claudia Voigt | 19.11.20 17:30:00 |
|---|---|---|

Zeilenabstand:  Mehrere 1.15 ze

| Seite 35: [71] hat formatiert | Claudia Voigt | 19.11.20 17:30:00 |
|---|---|---|

Schriftart: 10 Pt.

| Seite 35: [72] Formatiert | Claudia Voigt | 19.11.20 17:30:00 |
|---|---|---|

Zeilenabstand:  Mehrere 1.15 ze

| Seite 35: [73] hat formatiert | Claudia Voigt | 19.11.20 17:30:00 |
|---|---|---|

Schriftart: 10 Pt.

| Seite 35: [74] hat formatiert | Claudia Voigt | 19.11.20 17:30:00 |
|---|---|---|

Schriftart: 10 Pt.

| Seite 35: [75] hat formatiert | Claudia Voigt | 19.11.20 17:30:00 |
|---|---|---|

Schriftart: 10 Pt.

| Seite 35: [76] hat formatiert | Claudia Voigt | 19.11.20 17:30:00 |
|---|---|---|

Schriftart: 10 Pt.

| Seite 35: [77] Formatiert | Claudia Voigt | 19.11.20 17:30:00 |
|---|---|---|

Zeilenabstand:  Mehrere 1.15 ze

| Seite 35: [78] hat formatiert | Claudia Voigt | 19.11.20 17:30:00 |
|---|---|---|

Schriftart: 10 Pt.

| Seite 35: [79] hat formatiert | Claudia Voigt | 19.11.20 17:30:00 |
|---|---|---|

Schriftart: 10 Pt.

| Seite 35: [80] hat formatiert | Claudia Voigt | 19.11.20 17:30:00 |
|---|---|---|

Schriftart: 10 Pt.

| Seite 35: [81] hat formatiert | Claudia Voigt | 19.11.20 17:30:00 |
|---|---|---|

Schriftart: 10 Pt.

| Seite 35: [82] Formatiert | Claudia Voigt | 19.11.20 17:30:00 |
|---|---|---|

Zeilenabstand:  Mehrere 1.15 ze

| Seite 35: [83] hat formatiert | Claudia Voigt | 19.11.20 17:30:00 |
|---|---|---|

Schriftart: 10 Pt.

| Seite 35: [84] hat formatiert | Claudia Voigt | 19.11.20 17:30:00 |
|---|---|---|

Schriftart: 10 Pt.

| Seite 35: [85] hat formatiert | Claudia Voigt | 19.11.20 17:30:00 |
|---|---|---|

Schriftart: 10 Pt.

| Seite 35: [88] hat formatiert | Claudia Voigt | 19.11.20 17:30:00 |
|---|---|---|

Schriftart: 10 Pt.

| Seite 35: [89] hat formatiert | Claudia Voigt | 19.11.20 17:30:00 |
|---|---|---|

Schriftart: 10 Pt.

| Seite 35: [90] hat formatiert | Claudia Voigt | 19.11.20 17:30:00 |
|---|---|---|

Schriftart: 10 Pt.

| Seite 35: [91] Formatiert | Claudia Voigt | 19.11.20 17:30:00 |
|---|---|---|

Zeilenabstand:  Mehrere 1.15 ze

| Seite 35: [92] hat formatiert | Claudia Voigt | 19.11.20 17:30:00 |
|---|---|---|

Schriftart: 10 Pt.

| Seite 35: [93] hat formatiert | Claudia Voigt | 19.11.20 17:30:00 |
|---|---|---|

Schriftart: 10 Pt.

| Seite 35: [94] hat formatiert | Claudia Voigt | 19.11.20 17:30:00 |
|---|---|---|

Schriftart: 10 Pt.

| Seite 35: [95] hat formatiert | Claudia Voigt | 19.11.20 17:30:00 |
|---|---|---|

Schriftart: 10 Pt.

| Seite 35: [96] Formatiert | Claudia Voigt | 19.11.20 17:30:00 |
|---|---|---|

Zeilenabstand:  Mehrere 1.15 ze

| Seite 35: [97] hat formatiert | Claudia Voigt | 19.11.20 17:30:00 |
|---|---|---|

Schriftart: 10 Pt.

| Seite 35: [98] hat formatiert | Claudia Voigt | 19.11.20 17:30:00 |
|---|---|---|

Schriftart: 10 Pt.

| Seite 35: [99] hat formatiert | Claudia Voigt | 19.11.20 17:30:00 |
|---|---|---|

Schriftart: 10 Pt.

---

## Author Response (AR2)

Dear Mrs Ali,

Thank you for considering our manuscript entitled "Triple oxygen and isotope systematics of evaporation and mixing processes in a dynamic desert lake system" for publication in *Hydrology and Earth System Sciences*. We highly appreciate the constructive comments and further suggestions of the two reviewers. We followed almost all suggestions of both reviewers. In our opinion, the proposed splitting of the section on the "determination of the turbulence coefficient" does not lead to an improvement of the readability of the manuscript, and is better presented in context. Therefore, we have not implemented this. Please find attached a revised version of our contribution and a detailed point-by-point response to the reviewer's comments below.

Reviewer 1:

I congratulate the authors for a thorough revision of their paper, which now demonstrates the utility and benefits of triple isotope studies in arid environments. The methods improved and the paper clearly conveys important key messages. The only very minor suggestion I still have is that I would urge the authors to include the three specific objectives stated in the response to Reviewers also at the end of the introduction in the paper.

We highly appreciate the comments of Reviewer #1 and list now the three previously stated objectives in the introduction.

Reviewer 2:

General Comments

The authors have done a good job revising the manuscript, which has greatly improved the readability and replicability of the results. Nevertheless, there were a few instances of odd structuring (e.g. methods in results, results in discussion) that should be revised. The majority of this revision is minor clarification and rearranging text already within the manuscript. I have highlighted the suggested changes in the specific comments below.

We again appreciate the constructive comments and suggestions of Reviewer #2, which helped to finalize the manuscript for publication. We followed these suggestions in the revised version of the manuscript. Please, find below in black the comments of the reviewer, in blue our responses to the comments and how these comments are addressed in the revised manuscript.

Specific Comments

General comments throughout the manuscript: Throughout, suggest changing "about" to "approximately" for more formality. Check the presented precision of the data. Significance levels change throughout the manuscript (suggest keep to 1 decimal)

We changed "about" to "approximately", where we found it to be appropriate. We checked the precision. Presented significant levels are 1 digit for primary isotope parameters ($\delta^{17}O$, $\delta^{18}O$, $\delta^2H$) and no digits for secondary isotope parameters (d-excess, $^{17}O$-excess).

P1L28: "The GMWL describes the equilibrium…" There needs to be some context here. The GMWL described the equilibrium of deuterium and oxygen-18 (these isotopes are not mentioned above).

Sentence rephrased.

P2L31: suggest changing "from" to "using"

Done.

P2L33: I would take out groundwater recharge from here. The C-G model can be applied to pan evaporation (i.e. not continuous groundwater recharge).

We agree, the C-G model can also be applied to pan evaporation. However, in this case the 'groundwater recharge' is still a variable assuming a value of zero.

P2L37: Suggest adding a reference here.

Reference added.

P3L73: Are these annual average fluctuations? Or seasonal averages

These are 'mean seasonal values', i.e. average values of the respective season (Dec-Mar for austral summer, Jun-Sept for austral winter) averaged over the period since the installation of the meteorological station in 2015.

P3L74: Can you provide a value for "calm" as you did for "very windy"?

The wind speed is usually < 1 m/s in the morning. Information added.

P3L83: Suggest changing "higher" to "more enriched"

The water is more enriched in $^{18}O$ and thus comprises a higher $\delta^{18}O$ value. Thus, it's more correct using "higher".

P3L91: Suggest changing "rainy season" to "rainy season (summer)"

Done.

P5L133: Suggest revision to "The 600 ml pan dried up…"

Done.

P5L136: Swap the numbers to for the order listed (RH then temp)

Done.

P5L148: Why is T2 presented before T1? Suggest changing the order.

To keep the logical structure in the supplement (Terminology before showing results), we decided not to change the order.

P6L174: "relative humidity, h, normalized …"

Done.

P6L174: Do you mean that the RH at 1.5m is adjusted to the surface based on water surface temperature?

Yes.

P6L182: I would suggest having the definitions of both α's earlier with the equations. The causation is fine to keep here.

Done.

P6L185: Specify that "n" is the fractionation factor. Again, this would be better to be introduced with the equations above. Please change the location of the n superscript for consistency with the equation.

We removed n from the equations and introduce it now separately below.

P7L201: It is odd to switch from using R to denote isotopic composition and δ notation here. If R is different from the isotopic composition δ (i.e. not standardized, in concentration, etc) then it should be specified.

We replaced "composition" by "ratio" in the definition of R and added the formula relating R and δ.

P7L201: If I understand correctly for the later mixing models, you use this equation ($\delta X_{mix}$) for mixing the incoming water. If not, where is $\delta X_{mix}$ used? It might be better to use the RWS equation from T1 here if that is used for Figure 11.

This equation defines the mixing line shown in Figure 3, 4, and 10. The $R_{WS}$ equation defines recharge evaporation trajectories shown in Figure 3, 4, and 11.

P7L202: Remove the underline from Xmix

Done.

P7L205: Suggest changing to "..mixing processes are likely transient…"

Done.

P8L247-248: Increasing and decreasing with relative to what? Time periods?

Values of $\delta^{18}O$ increase with decreasing $^{17}O$-excess and d-excess, which is characteristic for evaporation. Sentence rephrased.

P9L262-268: Parts of this section seem more suited for the discussion than results. Additionally, the statement of similarities between vapour estimated by OIPC and measured comes before the values of OIPC are given.

The isotopic composition of the atmospheric vapor is a prerequisite for modelling the C-G evaporation trajectories, and not the focus of the manuscript. Therefore, we think it improves readability when the determination of individual parameters of the C-G equation is presented in individual subsections rather than distributing the information over the whole manuscript.

P9L272-273: Initial composition is better suited to the methods section rather than results.

This is the results of the measurement and thus belongs in the results section.

P9L276: No need to describe what the n value is again here.

We agree.

P9L277-281: This should be in the methods section to describe how you get the n value. The reader should know how you will get the n value before the results.

How to derive the turbulence coefficient is a method that is not established but rather developed within the manuscript. Thus, from our view it belongs in the results / discussion section.

P10L283: I think it would be more correct to say that the fit produces the X n value, suggesting higher than average turbulence (i.e. n < 0.5).

Sentence rephrased.

P10L283-286: The comparison to literature and explanation of differences is better given in the discussion than here.

We agree that this section includes results as well as discussion of the pan evaporation experiment. However, this subsection aims to determine the turbulence coefficient used for the C-G model, which is a prerequisite for applying the C-G model presented in the discussion. From our point of view, dividing in several subsection would reduce the readability of the manuscript.

P10L287: As the comparison of δ2HV, δ18OV, and d-excess v is important to understand here, the rationale for examining all three should be moved the methods section alongside the methods of obtaining n. Right now, the examination of δ2HV, δ18OV seem separate and as an afterthought to d-excess. It would help the flow of this section to present the fitting of δ2HV, δ18OV, and d-excess v, then present the sensitivity of d-excess fit n as it gives the most reasonable value (and by reasons described by Gonfiantini et al., 2020).

In fact, we used two methods to derive the turbulence coefficient. 1) Fitting the evaporation trajectory individually to $\delta^{18}O$ and $\delta^{2}H$ vs the fraction of remaining water. 2) Fitting the evaporation trajectory directly to d-excess vs the fraction of remaining water. The latter is advantageous due to its insensitivity to other model input parameters, especially $\delta^{18}O_V$. We rephrased these two paragraphs to clarify this.

P10L290: Figure 8 caption is in the incorrect order (should be d-excess, δ18O, the δ2H)

Thanks, changed.

P10L309-320: This discussion of where the C-G model is still deviating from the samples is better served in the discussion section 6.4, on the C-G- applicability.

We agree that this section combines results and discussion. However, for the reasons mentioned above, we decided to divide the sections by topic.

P11L327: Please provide the most relevant papers from within the study.

Done.

P12L355: Which figure is the envelope mixing referring to?

Fig. 4a. Added.

P12: Section 6.3. This whole section appears to be more methods for how to average the temperature and relative humidity of the mixing models than a discussion of the above-presented results. Most of this section should be moved to the methods, with the remaining going to the results (e.g. Section 5.1. presenting the conditions of the salar). This way the effect/change of the residence times can adequately be discussed in Section 6.4.

We agree and shifted this section in the Methods.

P13: Section 6.4. There is a great deal of this section that should be in the results as an evaluation of the C-G model robustness rather than in the discussion. The authors have improved on the presentation of mixing, though it restricted primarily to the supplementary material and is not referred to much. Additionally, the references to Fig 9 should be Fig 11. Further discussion of the differences/sensitivity of the C-G model would additionally help this section.

It is unclear what the reviewer is missing in our discussion. We have added two sentences on the relationship between $\delta^{18}O$ and salinity.

P14L423: I would suggest starting with a conclusion of the system before concluding findings of the C-G model, for consistency with the objectives.

We revised the conclusion and summarize now first the conclusions drawn from the triple oxygen isotope system, before evaluating the impact of hydrological dynamics on the C-G model.

P14L425: A requirement for what? A good estimation using the C-G model?

A requirement for the application of the C-G model. Added.

P14L434: Suggest changing "fall on top of each other" to "overlap"

Done.